# Exploring conformational equilibria of a heterodimeric ABC transporter

**M Hadi Timachi[1,2†], Cedric AJ Hutter[3†], Michael Hohl[3], Tufa Assafa[1,2], Simon Böhm[4], Anshumali Mittal[5], Markus A Seeger[3]\*, Enrica Bordignon[1,2]\***

[1]Faculty of Chemistry and Biochemistry, Ruhr-Universität Bochum, Bochum, Germany; [2]Department of Physics, Freie Universität Berlin, Berlin, Germany; [3]Institute of Medical Microbiology, University of Zurich, Zurich, Switzerland; [4]Laboratory of Physical Chemistry, ETH Zurich, Zurich, Switzerland; [5]Department of Biochemistry, University of Zurich, Zurich, Switzerland

**Abstract** ABC exporters pump substrates across the membrane by coupling ATP-driven movements of nucleotide binding domains (NBDs) to the transmembrane domains (TMDs), which switch between inward- and outward-facing (IF, OF) orientations. DEER measurements on the heterodimeric ABC exporter TM287/288 from *Thermotoga maritima*, which contains a non-canonical ATP binding site, revealed that in the presence of nucleotides the transporter exists in an IF/OF equilibrium. While ATP binding was sufficient to partially populate the OF state, nucleotide trapping in the pre- or post-hydrolytic state was required for a pronounced conformational shift. At physiologically high temperatures and in the absence of nucleotides, the NBDs disengage asymmetrically while the conformation of the TMDs remains unchanged. Nucleotide binding at the degenerate ATP site prevents complete NBD separation, a molecular feature differentiating heterodimeric from homodimeric ABC exporters. Our data suggest hydrolysis-independent closure of the NBD dimer, which is further stabilized as the consensus site nucleotide is committed to hydrolysis.

**\*For correspondence:** m.seeger@ imm.uzh.ch (MAS); enrica. bordignon@rub.de (EB)

[†]These authors contributed equally to this work

**Competing interests:** The authors declare that no competing interests exist.

## Introduction

ABC transporters are divided into importers, found exclusively in bacteria, and exporters, present in all phyla of life (*Davidson et al., 2008*). Transport processes mediated by more than forty human ABC exporters fulfil vital functions in our body as they translocate an extraordinarily wide range of cargoes such as lipids, peptides, ions and drugs across lipid bilayers. A number of severe hereditary diseases including cystic fibrosis and insulin secretion disorders such as neonatal diabetes and hyper-insulinism are directly linked to ABC transporter malfunction (CFTR and SUR1, respectively), under-pinning their paramount importance in human health (*Aittoniemi et al., 2009*; *Gadsby et al., 2006*). The human ABC exporters P-glycoprotein, ABCG2 and MRP1 act as multidrug efflux pumps in tumors and hamper effective cancer treatment (*Gottesman et al., 2016*). The same transporters have received increasing attention in past years, playing a central role in the absorption, distribution, metabolism and elimination of pharmaceuticals in the human body (*Nigam, 2015*). In addition, ABC exporters play a major physiological role in the transport of metabolites such as urate, glucuronides and N-lactoyl-amino acids (*Jansen et al., 2015*; *Krumpochova et al., 2012*; *Woodward et al., 2009*).

ABC exporters minimally consist of two transmembrane domains (TMDs) each containing six transmembrane helices protruding far into the cytoplasm and two nucleotide binding domains (NBDs), which undergo large conformational changes in response to ATP binding and hydrolysis (*Seeger and van Veen, 2009*). NBDs are called 'closed' when they sandwich two ATP molecules at

their head-to-tail dimer interface or 'open' when they are partially or completely disengaged from each other. The NBDs are coupled to the TMDs via two main and two accessory coupling helices (*Dawson and Locher, 2006*). When the NBDs are disengaged, the TMDs adopt an IF state (*Ward et al., 2007*). In contrast, the TMDs adopt an outward-facing or outward-occluded state when the NBDs are closed (*Dawson and Locher, 2006*; *Choudhury et al., 2014*). Transport across the membrane requires binding of a substrate to the inward-facing cavity, NBD closure and transition to the OF state and substrate release. The outward-occluded state is a possible intermediate in the cycle. An affinity switch is a prerequisite for uphill transport of substrates, namely substrates need to bind with higher affinity to the inward-facing transporter than to the outward-facing transporter (*Higgins and Linton, 2004*; *Sauna and Ambudkar, 2007*). Deviations from this general substrate pathway exist for ABCA4, which in fact imports N-retinylidene-phosphatidylethanolamine (i.e. an inverse affinity switch must exist) (*Quazi et al., 2012*) and for ABC exporters transporting substrates which are too large to be accommodated in the cavity as it is proposed for PglK transporting lipid-linked oligosaccharides (*Perez et al., 2015*).

The question concerning the origin of the power stroke for transport is a recurrent matter of debate in the ABC transporter field (*George and Jones, 2012*). In the prevalent ATP-switch model postulated by Higgins and co-workers, ATP binding leads to NBD closure and a concomitant switch from the IF to the OF state (*Higgins and Linton, 2004*). ATP hydrolysis then destabilizes the NBD dimer and the transporter reverts to its IF state with disengaged NBDs. In the ATP-switch model, ATP hydrolysis is not required for substrate release; the entire transport reaction of a substrate is coupled to ATP binding to the NBD sandwich dimer and the role of ATP hydrolysis is to recharge the machinery for the next ATP binding and substrate transport reaction. Detailed studies on ABCB1 have emphasized the importance of asymmetric occlusion of one nucleotide to trigger the affinity switch and thus substrate release, which challenged the ATP-switch model. Nucleotide occlusion is achieved by beryllium fluoride and vanadate trapping of ADP (representing the pre- and post-hydrolysis states, respectively), the substitution of the Walker B glutamate to glutamine or alanine (trapping a pre-transition state of the nucleotides) and in the presence of the slowly hydrolyzable analog ATPγS (*Sauna and Ambudkar, 2001*; *Sauna et al., 2006*, *2007*).

Crystal structures of ABC exporters are with a few notable exceptions in line with the ATP-switch model. When crystallized in the absence of nucleotides (apo), all ABC exporter structures depict IF states with partially or fully separated NBDs (*Ward et al., 2007*; *Hohl et al., 2014*; *Kodan et al., 2014*; *Lee et al., 2014*; *Li et al., 2014*). Since nucleotide-free conditions do not exist in living cells, apo structures represent very rare events of the transport cycle, i.e. the exchange of two nucleotides at the same time. Most ABC exporter structures crystallized in the presence of nucleotides exhibit closed NBDs and adopt outward-occluded or outward-facing configurations of the TMDs (*Dawson and Locher, 2006*; *Choudhury et al., 2014*). Exceptions are inward-facing ABCB10 (with two AMP-PCP bound) (*Shintre et al., 2013*) and TM287/288 (with one AMP-PNP bound to the degenerate site) (*Hohl et al., 2012*). In agreement with the ATP-switch model, the majority of the ABC exporters with closed NBDs have been crystallized with the ATP analogs AMP-PNP and ATPγS (*Ward et al., 2007*; *Choudhury et al., 2014*; *Dawson and Locher, 2007*; *Lin et al., 2015*). However, in the structures of Sav1866 and PglK, closed NBD dimers with bound ADP were observed (*Dawson and Locher, 2006*; *Perez et al., 2015*); according to the ATP-switch model these NBDs are expected to be separated. So far, no asymmetries in closed NBDs of full-length ABC exporter structures have been reported. Hence, structural evidence for asymmetric nucleotide occlusion as it has been proposed based on biochemical studies of ABCB1 is still missing. Accordingly, there is no crystallographic data in support of the constant contact model, which in contrast to the ATP-switch model posits that one of the nucleotide binding sites always remains bridged by at least one sandwiched nucleotide (*George and Jones, 2012*).

Crystal structures represent highly accurate snapshots of conformational states relevant for the transport cycle, but they do not report on the transporters' dynamics. Besides computational molecular dynamics simulations, pulsed electron paramagnetic resonance (EPR) techniques have been widely used as an experimental method to study frozen snapshots of conformational ensembles of ABC exporters. In particular, the homodimeric lipid A transporter MsbA has been studied in great detail by double electron electron resonance (DEER) (*Borbat et al., 2007*). The surprisingly large conformational changes between the apo and nucleotide-bound states as they were observed in the various MsbA crystal structures (*Ward et al., 2007*) could be confirmed by DEER in detergent

solution and in proteoliposomes (*Mittal et al., 2012*; *Zou et al., 2009*). Both binding of AMP-PNP-Mg and trapping of ADP-vanadate, representing the pre- and the post-hydrolytic state, respectively, were sufficient for NBD closure and complete transition to the outward-facing configuration of the TMDs, supporting the ATP-switch model's key notion that ATP binding and not ATP hydrolysis drives the IF to OF transition. In a recent study, single particle electron microscopy (EM) was used to delineate the conformational trajectory of MsbA (*Moeller et al., 2015*). In agreement with the crystal structures and DEER studies, large separation between the NBDs in the absence of nucleotides was observed. Interestingly, the method allowed for detection of poorly populated states and revealed that even in the absence of nucleotides a small fraction of the MsbA adopts an OF state. The fraction of outward-facing MsbA was increased up to 95% in the presence of ATP-Mg-vanadate, but in contrast to the DEER studies reached only 41% in the presence of AMP-PNP-Mg.

The ATP-switch model for ABC exporters has recently been challenged by a DEER study conducted on the heterodimeric ABC exporter BmrCD, claiming that in contrast to homodimeric transporters, BmrCD requires ATP hydrolysis for complete NBD closure and TMD reorientation (*Mishra et al., 2014*). A hallmark of BmrCD and many other heterodimeric ABC exporters is the so-called degenerate ATP binding site, which binds ATP tightly but is catalytically impaired (*Basso et al., 2003*; *Procko et al., 2006*; *Tsai et al., 2010*). TM287/288 from the thermophilic bacterium *Thermotoga maritima* is currently the only heterodimeric ABC exporter encompassing a degenerate nucleotide binding site for which crystal structures are available. The TM287/288 structures were solved in the presence of AMP-PNP-Mg and in the apo state, respectively (*Hohl et al., 2012*, *2014*). Both structures are inward-facing and exhibit differences mainly at the D-loops, which were shown to establish functionally crucial cross-talk between the asymmetric binding sites (*Hohl et al., 2014*; *Furman et al., 2013*; *Grossmann et al., 2014*). In contrast to ABC exporters comprising two consensus sites, the NBDs of TM287/288 remain in contact mainly via the degenerate site D-loop even in the absence of nucleotides (*Hohl et al., 2014*). A subnanometer-resolution cryo-EM structure of the heterodimeric ABC exporter TmrAB from *Thermus thermophilus* determined in the absence of nucleotides is in support of this notion (*Kim et al., 2015*). DEER measurements on TM287/288 in detergent solution and proteoliposomes, in the absence of nucleotides and in the presence of AMP-PNP-Mg, were in agreement with the corresponding crystal structures, showing an inward-facing TMD domain and NBDs in partial contact. AMP-PNP-Mg was shown to be insufficient to fully close the NBDs and to support the transition to the OF state (*Hohl et al., 2014*).

Here we investigate the complete conformational cycle of the heterodimeric ABC exporter TM287/288 studying the dynamic consequences of nucleotides and nucleotide analogs added at saturating concentrations to the wildtype transporter and to the catalytically inactive E517Q[TM288] (E-to-Q) mutant. DEER measurements performed with ATP in the absence of the co-factor magnesium revealed that a fraction of transporters switched to the OF state without ATP hydrolysis. Measurements performed under the same experimental conditions with BmrCD and MsbA highlight analogies and differences between the energy landscape of these ABC exporters. Furthermore, it is demonstrated that, in the absence of nucleotides, the NBDs of TM287/288 asymmetrically disengage upon heating to a physiological temperature of 80°C in a reversible fashion. In this state, the conformational dynamics of the NBDs are not communicated to the TMDs, resulting in decoupled movement of the NBDs from the rest of the protein. Due to the stabilization of cross-NBD contacts mediated by binding of a nucleotide to the degenerate ATP binding site, NBD separation at high temperature does not occur in the presence of nucleotides.

Our findings show that the energy landscape of TM287/288 is different from that of BmrCD and MsbA. The recently proposed diverging conformational cycle for heterodimeric ABC exporters, which seemingly requires ATP hydrolysis as a power stroke to progress to the OF state, is called into question.

## Results

### Conformational switch to the OF state in wildtype TM287/288 by ATP-Mg and vanadate trapping

Six spin-labeled pairs were introduced into cys-less TM287/288 (called wildtype TM287/288 for simplicity): two pairs in the NBDs to monitor movements at the consensus and degenerate ATPase sites,

two in the intracellular part of the TMDs and two in the extracellular part of the TMDs. Simulations performed with a rotamer library of spin-labeled side chains available in the software MMM (*Polyhach et al., 2011*) using the apo structure of TM287/288 and a homology model based on Sav1866 indicated that the six pairs allow monitoring of the conformational changes propagated from the NBDs to the TMDs (*Figure 1* and *Figure 1—figure supplement 1*). Four out of these six pairs were already used in a previous study (*Hohl et al., 2014*) but investigated only under apo and AMP-PNP-Mg conditions. Here, we investigated a comprehensive set of ATP analogs and experimental conditions to trigger the conformational transitions in this ABC exporter (*Figure 2* and *Figure 2—figure supplement 3*). Nucleotides were used at a concentration of 2.5 mM together with 2.5 mM $MgCl_2$ (indicated as nucleotide-Mg) throughout the study. To address the effect of ATP binding alone on the conformational transition, we also used ATP (2.5 and 14 mM) in the presence of 2.5 mM EDTA to chelate the $Mg^{2+}$ ions. All spin-labeled mutants (spin labeling efficiency >70%) were shown to retain robust ATPase activity (>90%) (*Table 1*). Spin-labeled mutants as well as wild-type TM287/288 were reconstituted into proteoliposomes and stimulation of ATP hydrolysis in the presence of 50 μM, 100 μM and 150 μM Hoechst 33342 was determined (*Figure 1—figure supplement 2*). Data were normalized to the ATPase activity of reconstituted wildtype TM287/288 in the absence of drug. All spin-labeled mutants exhibited a robust ATPase stimulation upon addition of Hoechst 33342 which was maximal at 50 μM or 100 μM Hoechst 33342 depending on the mutant investigated. As in other heterodimeric multidrug ABC exporters (*Hürlimann et al., 2016*) the curves were bell-shaped, i.e. the ATPase activity declined between 100 μM and 150 μM Hoechst 33342. The maximal ATPase stimulation was less pronounced for the extracellular pairs and for the intracellular pair 131[TM288]/248[TM288], indicating that spin-labeling at these positions changes the properties of TM287/288 in terms of drug binding or communication between the TMDs and the NBDs underlying stimulated ATPase activity in the presence of drugs. Despite these modest differences, all spin-labeled mutants were functional and retained the ability to sense drug binding.

The six interspin distances measured in the apo state of the transporter are in agreement with the crystallized IF conformation (*Figure 1* and *Figure 1—figure supplement 1*) based on the MMM simulations, which were previously shown to provide a 3.5–4 Å root mean square deviation (rmsd) between experimental and calculated mean distances (*Jeschke, 2013*). In fact, the tolerance between simulated mean distances to the experimental ones in the six pairs agree within this rmsd when using the 2013 MMM library; the 2015 library gives slightly worse results (especially for pairs 150–295 and 350–475) (*Figure 1* and *Figure 1—figure supplement 1*). Addition of ATP-Mg in the presence of vanadate (ATP-Vi-Mg) induced distinct distance changes in all six pairs compared to the apo state (*Figure 2*, magenta versus cyan lines). ADP trapping by vanadate, which is expected to occur in the catalytically active consensus site, switched the great majority of the transporters to a conformation strongly resembling the homology model based on the outward-facing structure of Sav1866 (*Figure 1*). With respect to the apo state, the NBDs and intracellular TMD pairs showed a decrease in interspin distance. Concomitantly, the two pairs in the extracellular region showed an increase of the mean distances. Intriguingly, a large width of the distance distribution was obtained in the extracellular region, encompassing the distances already present in the apo state (*Figure 2*), possibly highlighting structural heterogeneities. The mean distances obtained in this state are within an acceptable level of agreement with those predicted by MMM on the OF homology model (*Figure 1—figure supplement 1*). It should be noted that in this case the errors caused by the coarse-grained library approach add to the intrinsic errors produced by homology modeling in terms of side-chain arrangements, which influence the number of spin label rotamers calculated, and as a consequence thereof the distance distributions.

A high fraction of OF states was also obtained by incubating spin-labeled TM287/288 with ATP-Mg for 20 s at 25°C (*Figure 2*, red lines). Since TM287/288 stems from the hyperthermophilic bacterium *Thermotoga maritima*, ATP hydrolysis at 25°C is comparatively slow (turnover rate of 11 ATP min$^{-1}$, *Table 1*) and thereby resembles the experimental condition of ATP hydrolysis inhibited by vanadate at 50°C (turnover rate of 22 ATP min$^{-1}$, *Table 1*). Inhibition of ATPase activity by vanadate was measured over a range of vanadate concentrations at 50°C (*Figure 3B*). Based on fitting and calculations as outlined in the material and methods, vanadate has a $K_i$ of 75.1 ± 11.4 nM (*Table 2*). Of note, at 1 mM vanadate concentration the residual ATPase activity was 8% (*Figure 3B*). This indicates that the vanadate-trapped state of TM287/288 is not very stable and a fraction of transporters run through the transport cycle.

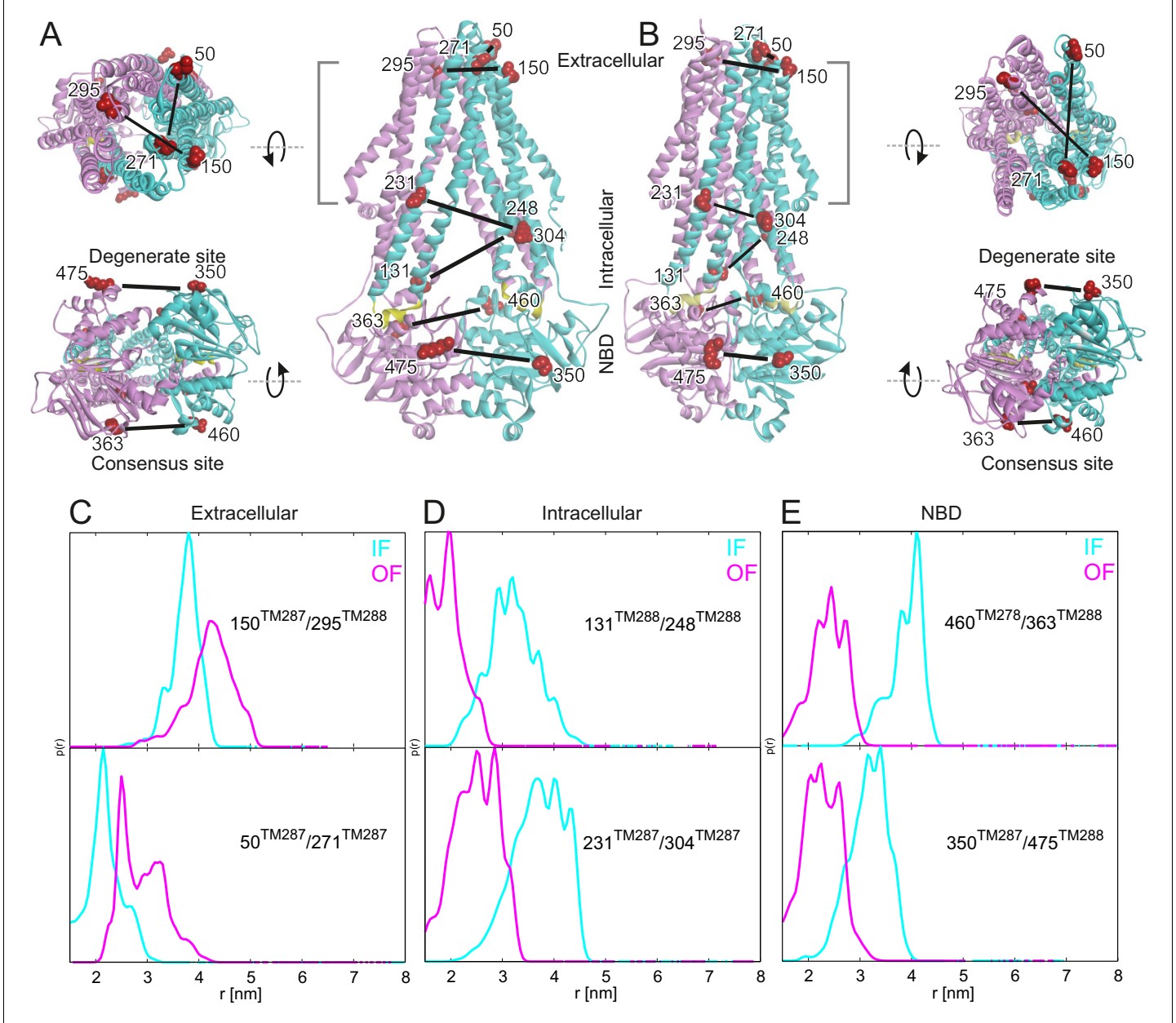

**Figure 1.** Labeling sites and putative conformational switch. Schematic of spin-labeled sites in the extracellular, intracellular and NBD regions of TM287/288 in (**A**) the inward-facing apo crystal structure (PDB: 4Q4H) and in (**B**) the outward-facing homology model based on the Sav1866 crystal structure (PDB: 2HYD). TM287 is colored in cyan and TM288 in pink. (**C–E**) Simulations of the distance distribution probabilities for the six spin-labeled double mutants in the IF (cyan) and OF (magenta) states represented in panels A and B. The ambient temperature MTSL rotamer library in MMM2015 was used. Comparison with the experimental data and with a previous version of the MTSL library are presented in **Figure 1—figure supplement 1**.

The following figure supplements are available for figure 1:

**Figure supplement 1.** Comparison between simulated and experimental distance distributions.

**Figure supplement 2.** Hoechst 33342 stimulated ATPase activities of wildtype TM287/288 and spin-labeled mutants reconstituted into proteoliposomes.

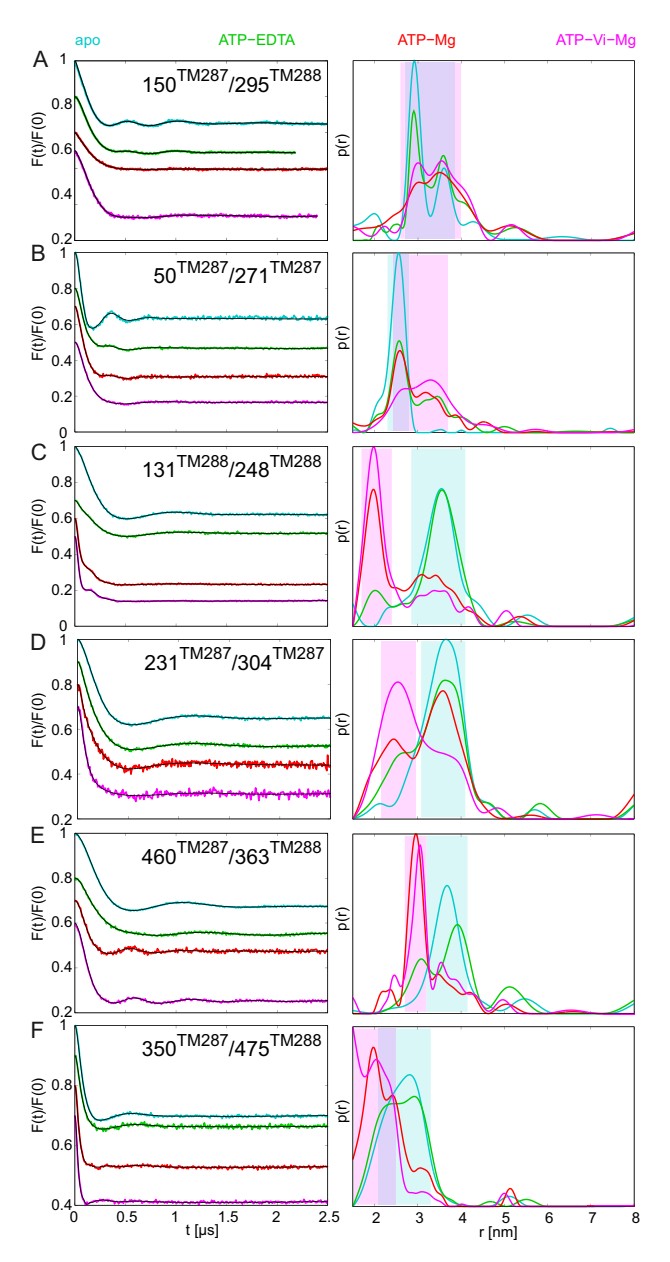

**Figure 2.** DEER analysis of spin-labeled TM287/288. Q-band background-corrected DEER traces [F(t)/F(0)] with fitted distribution function (left) and corresponding distance distribution (right) calculated using DeerAnalysis2015 and normalized by area. Traces are shown for the six spin-labeled pairs in the absence of nucleotides (apo, cyan), ATP-EDTA (green), ATP-Mg incubated for 20 s (red) and ATP-Vi-Mg (magenta). (**A** and **B**) Extracellular pairs. (**C** and **D**) Intracellular pairs. (**E**) NBD pair 460TM287/363TM288 monitoring the consensus site. (**F**) NBD pair 350TM287/ 475TM288 monitoring the degenerate site. Transparent cyan and magenta rectangles outline the range of experimental distances characteristic of the IF and OF conformations, respectively. All primary DEER data can be found in *Figure 2—figure supplement 1*. The DEER data of the apo states presented in panels **A, C, E, F** are taken from (*Hohl et al., 2014*). Traces detected after incubation with 14 mM ATP and 2.5 mM EDTA are shown in *Figure 2—figure supplement 4*.

The following figure supplements are available for figure 2:

**Figure supplement 1.** Primary DEER traces of spin-labeled pairs in wildtype TM287/288.

**Figure supplement 2.** DEER analysis of TM287/288 in proteoliposomes.

*Figure 2 continued on next page*

*Figure 2 continued*

**Figure supplement 3.** DEER analysis of spin-labeled pairs in wildtype TM287/288 with additional nucleotide analogs and ADP-Mg.
**Figure supplement 4.** DEER analysis of wildtype TM287/288 in the presence of 2.5 mM or 14 mM ATP and 2.5 mM EDTA.

In agreement with this notion, we consistently observed a small fraction of transporters showing DEER distances corresponding to the IF state under vanadate-trapped conditions (*Figure 2*).

TM287/288 spin-labeled at the intracellular region (pair 131$^{TM288}$/248$^{TM288}$) was reconstituted into liposomes made of *E.coli* polar lipids and egg phosphatidylcholine. In line with the measurements performed with the detergent-solubilized transporter, membrane-embedded TM287/288 confirmed the switch to the OF state upon vanadate trapping as well as in the presence of ATP-Mg (i.e. under hydrolyzing conditions) (*Figure 2—figure supplement 2*). However, in liposomes the OF state was found to be less populated than in detergent. This discrepancy is likely owing to the directionality of the reconstitution; transporters having NBDs inside the lumen of the proteoliposome are inaccessible for ATP and consequently cannot contribute to the OF state population. The less pronounced shift to the OF state could be additionally due to the missing substrate in the assay, which might modulate the energy landscape of the reconstituted protein.

## Conformational dynamics of TM287/288 induced by AMP-PNP-Mg, ATPγS-Mg and ADP-Mg

In agreement with our previous study (*Hohl et al., 2014*), addition of the nucleotide analog AMP-PNP-Mg resulted only in modest conformational changes, most likely originating from the reorientation of the spin-labeled side chains (*Figure 2—figure supplement 3*). Thus, AMP-PNP-Mg binding does not support the conformational switch to the OF state in TM287/288, neither in detergent nor in proteoliposomes (*Hohl et al., 2014*) (*Figure 2—figure supplements 2–3*). By contrast, when the ATP analog ATPγS-Mg was added, a fraction of transporters was found to adopt the OF state, while a detectable fraction stayed in an apo-like conformation (*Figure 2—figure supplement 3*). Hence, there is a clear difference between the two broadly used ATP analogs AMP-PNP-Mg and ATPγS-Mg in terms of their ability to switch the conformation of TM287/288.

Direct incubation with the hydrolysis product ADP-Mg prevalently populated an IF conformation of the transporter (*Figure 2—figure supplement 3*, orange lines). Interestingly, the interspin distances of the ADP-Mg bound transporter in the intracellular pair 131$^{TM288}$/248$^{TM288}$ are found to be similar to the characteristic IF state in the presence of AMP-PNP-Mg, which is distinguishable from the apo state. Furthermore, in the two NBD pairs the distribution of distances shows some OF state contribution.

Nucleotide binding and hydrolysis follow a complex mechanism in heterodimeric ABC exporters, due to the existence of a degenerate and a consensus site in the NBDs. Both sites bind nucleotides with different affinities, but ATP hydrolysis occurs mainly if not exclusively at the consensus site. In addition, the ATP binding sites are allosterically coupled (*Hohl et al., 2014*; *Grossmann et al., 2014*). To study the apparent affinities of the nucleotide analogs and ADP, the ATPase activity of TM287/288 was measured in the presence of 0.5 mM ATP at increasing nucleotide concentration (*Figure 3B and D*). Assuming that changes of the ATPase activity are mainly caused by competition at the consensus site, the data were fitted with a single site hyperbolic function (see materials and methods) to determine an IC$_{50}$ of these compounds. Using the relation IC$_{50}$=K$_i$(1+[S]/K$_m$), K$_i$ values for the respective nucleotides were determined (*Table 2*). To conduct this calculation, an accurate K$_m$ for ATP hydrolysis at the respective temperature was measured (*Figure 3A and C*). When the ATPase assay was performed at 50°C, we determined K$_i$ values of 0.0439 ± 0.0062 μM, 0.246 ± 0.042 μM and 10.4 ± 2.9 μM for ATPγS, AMP-PNP and ADP, respectively (*Table 2*). When ATPase activities were determined at 25°C (i.e. the temperature at which spin-labeled TM287/288

**Table 1.** ATPase activities in detergent.

| Protein | Nucleotide | Temperature [°C] | ATPase activity [nmol Pi/min/mg protein] | Turnover per transporter [min$^{-1}$] | % of wildtype |
|---|---|---|---|---|---|
| TM287/288 | wildtype | | | | |
| | ATP - Mg | 50 | 2141 ± 67 | 284 | - |
| | | 25 | 86.1 ± 2.5 | 11.4 | - |
| | ATP – Mg +1 mM vanadate | 50 | 165 ± 20 | 21.9 | 7.71 |
| | ATP – Mg +2.5 mM EDTA | 50 | <0.1 | <0.01 | <0.005 |
| | AMP-PNP - Mg | 25 | <0.1 | <0.01 | <0.005 |
| | E517Q$^{TM288}$ | ATP - Mg | 25 | 0.165 ± 0.015 | 0.0219 | 0.192 |
| Spin-labeled TM287/288 | 350$^{TM287}$/475$^{TM288}$ | ATP - Mg | 50 | - | - | 193* |
| | 460$^{TM287}$/363$^{TM288}$ | ATP - Mg | 50 | - | - | 212* |
| | 131$^{TM288}$/248$^{TM288}$ | ATP - Mg | 50 | - | - | 156* |
| | 231$^{TM287}$/304$^{TM287}$ | ATP - Mg | 50 | - | - | 146* |
| | 150$^{TM287}$/295$^{TM288}$ | ATP - Mg | 50 | - | - | 112* |
| | 50$^{TM287}$/271$^{TM287}$ | ATP - Mg | 50 | - | - | 73* |
| BmrCD | wildtype | ATP - Mg | 25 | 22.9 ± 0.5 | 3.24 | - |
| | E592Q$^{BmrD}$ | ATP - Mg | 25 | 0.633 ± 0.053 | 0.0896 | 2.76 |
| MsbA | wildtype | ATP - Mg | 30 | 135 ± 9 | 18.4 | - |
| | 561$^{MsbA}$ | ATP - Mg | 30 | 122 ± 17 | 16.6 | 90.4 |

* values are given in respect to an internal wildtype control in each measurement.

were pre-incubated before flash-freezing), the corresponding $K_i$ values were 0.204 ± 0.032 μM, 1.17 ± 0.51 μM and 3.28 ± 0.44 μM for ATPγS, AMP-PNP and ADP, respectively (*Figure 3C*). These measurements revealed that at both temperatures ATPγS is the strongest inhibitor of ATP hydrolysis. It is well established that ATPγS is slowly hydrolyzed by ATPases, as for example the $F_1F_o$-ATPase (*Turina and Capaldi, 1994*). In ABCB1 the rate of hydrolysis of ATPγS was 0.4% of that determined for ATP (*Siarheyeva et al., 2010*). It is reasonable to assume that TM287/288 is as well capable of slow ATPγS hydrolysis. In addition, we tested whether TM287/288 hydrolyzes AMP-PNP. In contrast to ATPγS, hydrolysis of AMP-PNP releases $P_i$, which can be detected by the molybdate/malachite green method. We found that TM287/288 is unable to cleave AMP-PNP within the sensitivity of our assay (<0.01 $P_i$ min$^{-1}$) (*Table 1*).

## Populating an OF conformation without ATP hydrolysis

Up to this point, it is tempting to suggest that ATP hydrolysis and not nucleotide binding per se is required for the transition to the OF state, i.e. the switch was obtained under turnover conditions (ATP-Mg), with vanadate trapping as well as with the slowly hydrolyzable ATPγS but not with the non-cleavable AMP-PNP. To test whether the transition to the OF state also occurs in the absence of ATP hydrolysis, 2.5 mM EDTA was added to chelate any remaining $Mg^{2+}$. Although our ATPase assay would have permitted a reliable detection of a residual turnover of as little as 0.01 ATP min$^{-1}$ (*Table 1*), we did not observe any detectable ATPase activity under these conditions. We therefore concluded that the presence of $Mg^{2+}$ is an absolute requirement for ATPase activity of TM287/288 and that 2.5 mM EDTA is sufficient to completely chelate $Mg^{2+}$. DEER measurements on samples pre-incubated with ATP-EDTA revealed that a fraction of transporters underwent the conformational transition to the OF state, while the majority stayed in an apo-like IF conformation (*Figure 2*, green lines). Thus, in TM287/288 the transition to the OF state does not strictly require ATP hydrolysis. Instead, in agreement with the ATP switch model (*Higgins and Linton, 2004*) solely ATP binding (even in the complete absence of magnesium) is sufficient to induce the transition. However, it should be emphasized that in contrast to vanadate trapping, the IF-OF equilibrium was only partially

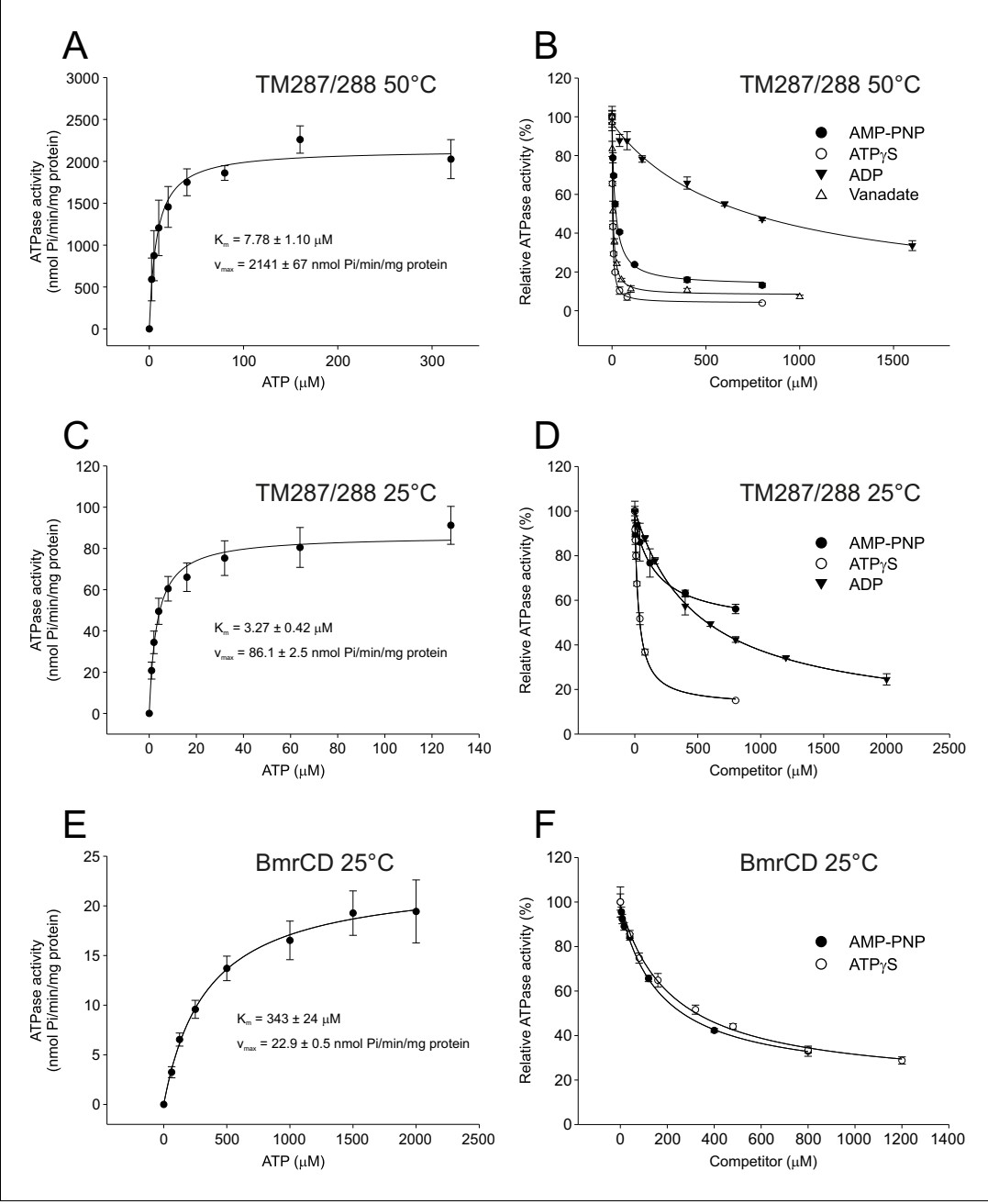

**Figure 3.** Inhibition of ATPase activity of TM287/288 and BmrCD by vanadate and nucleotides. $K_m$ and $v_{max}$ values for ATP hydrolysis by TM287/288 at 50°C (**A**) and 25°C (**C**) and BmrCD at 25°C (**E**) were determined by measuring ATPase activities at increasing ATP concentrations. Inhibition of ATP hydrolysis of TM287/288 was determined in the presence of increasing concentrations of vanadate at 50°C (**B**) or AMP-PNP, ATPγS and ADP at 50°C (**B**) or 25°C (**D**). Inhibition of ATP hydrolysis of BmrCD by AMP-PNP and ATPγS was determined at 25°C (**F**). In the inhibition assays, 500 μM and 2500 μM ATP were used for TM287/288 and BmrCD, respectively. The curves were fitted with a hyperbolic decay function to obtain $IC_{50}$ values, which were used together with the corresponding $K_m$ to calculate $K_i$ (**Table 2**). The error bars of the measurement points are standard deviations of three technical replicates.

shifted to the OF state in the presence of ATP-EDTA. Due to a lack of ATP hydrolysis, the $K_m$ of ATP in the absence of $Mg^{2+}$ ions could not be determined, and 2.5 mM ATP-EDTA may in fact represent a non-saturating nucleotide concentration. Therefore, DEER traces for three selected pairs were also

determined in the presence of 14 mM ATP and 2.5 mM EDTA. No significant differences were observed between distance distributions obtained at 2.5 mM or 14 mM ATP under $Mg^{2+}$-free conditions, showing that the results report on IF-OF equilibria obtained at a saturating nucleotide concentration (*Figure 2—figure supplement 4*).

## Conformational equilibria explored in ATPase deficient E-to-Q mutant of TM287/288

The same six pairs were studied in the TM287/288_E517Q$^{TM288}$ mutant (called E-to-Q mutant below), which has the consensus site Walker B glutamate substituted by a glutamine. The ATPase turnover rate of the E-to-Q mutant was 0.022 ATP $min^{-1}$ at 25°C (*Table 1*). This residual activity is unlikely to arise from the degenerate site, because the substitution of the non-canonical Walker B aspartate of the degenerate site with asparagine in addition to the E-to-Q mutation of the consensus site did not further decrease the ATPase activity (data not shown). The spin-labeled E-to-Q samples were incubated with ATP-Mg for 10 min at 25°C prior to freezing, therefore the ATP turnover amounted to 0.22 per transporter. In other words, at least 78% of the transporters have never undergone a hydrolysis cycle during the incubation period. The apo states in the E-to-Q mutant showed interspin distances in all six pairs, which were almost superimposable to those obtained in the wild-type transporter, indicating that the transporter was structurally unaffected by the mutation (*Figure 4*, cyan lines to be compared with *Figure 2*). Addition of ATP-Mg resulted in the almost complete switch of the transporters to the OF state (*Figure 4*, red lines). Since the great majority of transporters have not undergone a single ATP hydrolysis reaction prior to sample freezing, we can conclude that ATP hydrolysis is not necessary for the conformational switch in the E-to-Q mutant. Rather, the E-to-Q mutation stalls the ATP hydrolysis reaction in the pre-hydrolytic state. Notably, addition of ATP-EDTA to the transporter carrying the E-to-Q mutation also resulted in an almost complete switch from the IF to the OF state (*Figure 4*, green lines). This finding is in line with a study on SUR1 carrying the E-to-Q mutation, which could be more efficiently switched to the OF state by ATP-EDTA than the wildtype protein. In SUR1, this finding was interpreted to be caused by the removal of the negatively charged glutamate from the consensus site, leading to an increased nucleotide affinity (*Ortiz et al., 2013*). However, this explanation does not hold true for TM287/288 because DEER traces of wildtype TM287/288 remained unchanged regardless whether 2.5 mM or 14 mM ATP were added together with 2.5 mM EDTA, meaning that all our measurements were conducted at saturating nucleotide concentrations. Our data therefore suggest that the E-to-Q mutant has an altered energy profile, resulting in a higher overall propensity to adopt the OF state. In agreement with this notion, incubation with ATPγS-Mg also resulted in a larger fraction of OF states in equilibrium with an apo-like state as compared to the wildtype transporter. Again, ATPγS-Mg was added at saturating concentration (>12000 fold over $K_i$ of ATPγS-Mg in wildtype TM287/288, see *Table 2*), and the observed differences cannot be explained by an increased affinity of ATPγS-Mg

**Table 2.** $K_i$ determination.

| Protein | Temperature [°C] | $K_m$ for ATP [μm][*] | Competitor | $IC_{50}$ [μm][*] | $K_i$ [μm][†] |
|---|---|---|---|---|---|
| TM287/288 | 50 | 7.78 ± 1.10 | AMP-PNP | 16.1 ± 1.6 | 0.246 ± 0.042 |
| | | | ATPγS | 2.86 ± 0.08 | 0.0439 ± 0.0062 |
| | | | ADP | 681 ± 160 | 10.4 ± 2.9 |
| | | | Vanadate | 4.90 ± 0.29 | 0.0751 ± 0.0114 |
| | 25 | 3.27 ± 0.42 | AMP-PNP | 179 ± 75 | 1.17 ± 0.51 |
| | | | ATPγS | 31.4 ± 2.8 | 0.204 ± 0.032 |
| | | | ADP | 505 ± 21 | 3.28 ± 0.44 |
| BmrCD | 25 | 343 ± 24 | AMP-PNP | 174 ± 24 | 21.0 ± 3.2 |
| | | | ATPγS | 221 ± 24 | 26.7 ± 3.3 |

[*] $K_m$ and $IC_{50}$ values and standard errors were obtained from fits shown in *Figure 3*.

[†] $K_i$ values and standard errors were calculated based on the given $K_m$ and $IC_{50}$ values as described in the materials and methods.

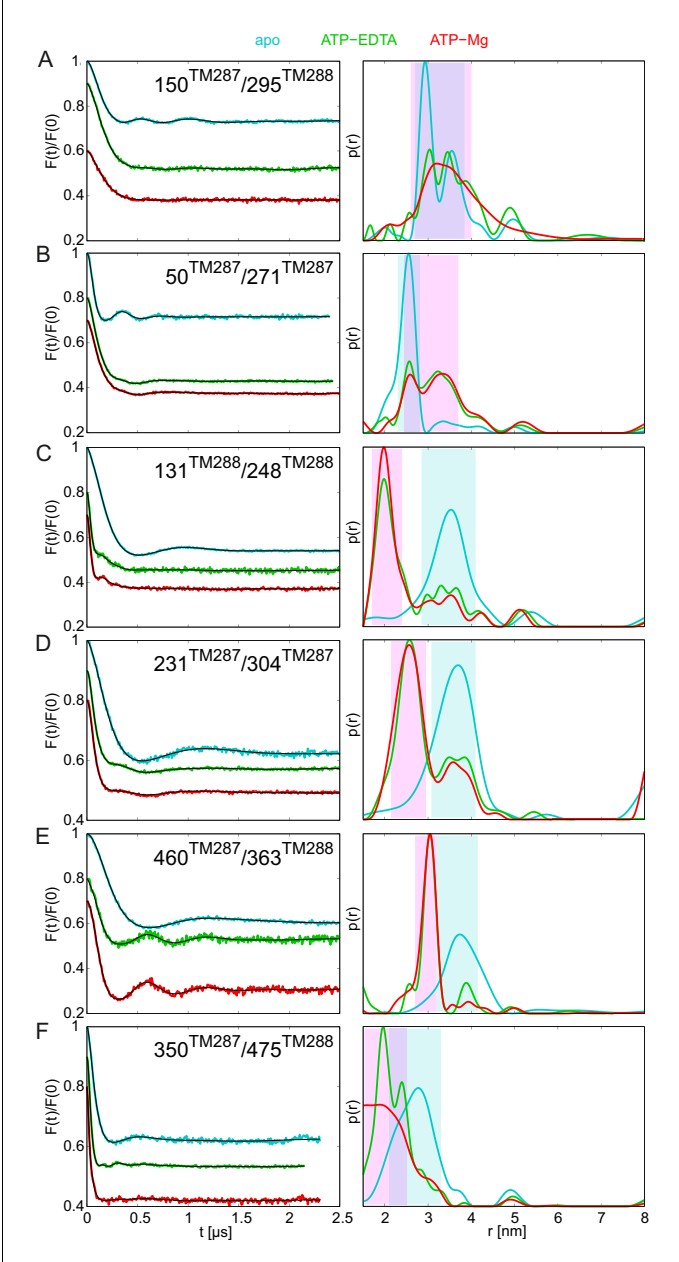

**Figure 4.** DEER analysis of spin-labeled TM287/288 carrying the E-to-Q mutation. Q-band background-corrected DEER traces [F(t)/F(0)] with fitted distribution function (left) and corresponding distance distribution (right) calculated using DeerAnalysis2015 and normalized by area. Traces are shown for the six spin-labeled pairs engineered in TM287/288 in the E-to-Q mutant in the absence of nucleotides (apo, cyan), ATP-EDTA (green) and ATP-Mg (red). (**A** and **B**) Extracellular pairs. (**C** and **D**) Intracellular pairs. (**E**) NBD pair $460^{TM287}/363^{TM288}$ monitoring the consensus site. (**F**) NBD pair $350^{TM287}/475^{TM288}$ monitoring the degenerate site. Transparent cyan and magenta rectangles outline the experimental distance range characteristic for the IF and OF conformations, respectively. All primary DEER data can be found in *Figure 4—figure supplement 1*.

The following figure supplements are available for figure 4:

**Figure supplement 1.** Primary DEER traces of spin-labeled pairs in TM287/288 carrying the E-to-Q mutation.

**Figure supplement 2.** DEER analysis of spin-labeled pairs in TM287/288 carrying the E-to-Q substitution with additional nucleotide analogs.

for the E-to-Q mutant. In addition to an altered energy profile, hydrolysis of ATPγS-Mg is further slowed down in the E-to-Q mutant relative to the wildtype transporter thereby prolonging the OF state lifetime (*Figure 4—figure supplement 2*, black lines).

AMP-PNP-Mg remained the poorest ATP analog in terms of its ability to facilitate the conformational switch of the E-to-Q mutant (*Figure 4—figure supplement 2*, purple lines).

Owing to the facilitated transition to the OF state in the E-to-Q mutant, IF/OF equilibria were observed for some spin-label pairs which were not apparent in the wildtype transporter. In the intracellular region, possibly a small OF-like distance peak can be identified with AMP-PNP-Mg in the

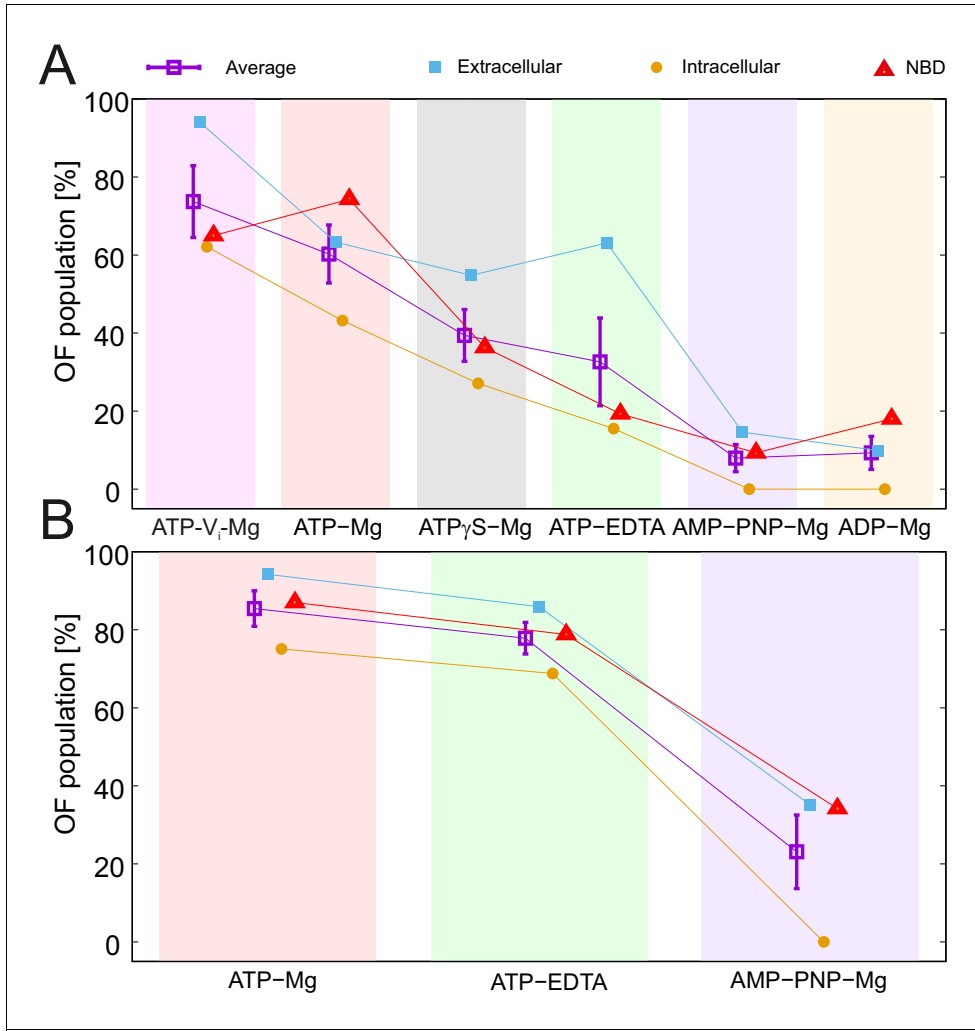

**Figure 5.** Nucleotide ranking according to the ability to populate the OF conformation. Three spin-labeled pairs of TM287/288 representing the extracellular region (150[TM287]/295[TM288]), intracellular region (131[TM288]/248[TM288]) and the NBDs (460[TM287]/363[TM288]) were used for the analysis of the IF/OF populations in the wildtype transporter (**A**) and the E-to-Q mutant (**B**). The percentage of the OF population was calculated using the relative area under the model-based two Gaussian distance distribution performed using DeerAnalysis2015. The purple squares with error bars show the average ability of each nucleotide to stabilize the conformational switch. Background-corrected DEER traces and corresponding distance distributions are shown in *Figure 5—figure supplement 1*.

The following source data and figure supplement are available for figure 5:

**Source data 1.** Parameters of the two-Gaussian fit.
**Figure supplement 1.** Two-Gaussian fit DEER analysis of three selected spin-labeled pairs of TM287/288.

pair 231$^{TM287}$/304$^{TM287}$, and in the extracellular region (pair 50$^{TM287}$/271$^{TM287}$) an indication of a broad distance which could be related to an OF-like state was detected (see also Gaussian analysis in *Figure 5—figure supplement 1*). Surprisingly, when monitoring the response of the NBDs, we found a relatively large fraction of the short distance peak (*Figure 5—figure supplement 1*) reporting the closure of the consensus site (pair 460$^{TM287}$/363$^{TM288}$) (*Figure 4—figure supplement 2E*, purple lines). In contrast, the degenerate site (pair 350$^{TM287}$/475$^{TM288}$) did not show changes in the distance distribution with AMP-PNP-Mg, suggesting an asymmetric closure of the NBDs with AMP-PNP-Mg. This unexpected finding seemingly contradicts the TM287/288 structure, which features asymmetric AMP-PNP-Mg binding to the degenerate site only. However, we wish to emphasize that in the inward-facing structure of TM287/288, AMP-PNP-Mg does not reach over to the ABC signature motif of the opposite NBD. Hence, the ABC signature motif of the degenerate site, which in fact deviates from the consensus LSGGQ motif, appears to be incapable of establishing a contact to AMP-PNP-Mg required to support complete NBD closure.

## A ranking of nucleotides with respect to the IF-OF transition

All nucleotides and nucleotide analogs used in this study were added at saturating conditions well above their $K_i$ values (*Table 2*). This also holds true for ATP-EDTA, which did not exhibit a change in distance distribution by increasing the ATP concentration from 2.5 mM to 14 mM (*Figure 2—figure supplement 4*). Therefore, the IF/OF fractions detected in the sample reflect the average life time that the nucleotide-bound transporter spends in either conformation. To rank the nucleotides in terms of their ability to support the transition to the OF conformation, we used three representative pairs in the wildtype transporter and the E-to-Q mutant: one in the extracellular region (50$^{TM287}$/271$^{TM287}$), one in the intracellular region (131$^{TM288}$/248$^{TM288}$) of the TMDs, and one in the NBDs (460$^{TM287}$/363$^{TM288}$) for which the distance peaks corresponding to the two states were distinguishable. A two-Gaussian fit of the DEER traces was performed to obtain the area of the corresponding IF and OF distance peaks (*Figure 5—source data 1*). The rank of nucleotides with respect to increasing fractions of the OF populations (*Figure 5A*) was: ADP-Mg ≈ AMP-PNP-Mg < ATP-EDTA < ATPγS-Mg < ATP-Mg < ATP-Vi-Mg. The order is similar for the wildtype and E-to-Q mutant transporter. However, the OF state is in general more populated in the E-to-Q mutant, which is best explained by an alteration of the energy profile caused by the mutation. This notion is best appreciated for ATP-EDTA and ATP-Mg, which could trigger an almost complete switch in the molecular ensemble of the E-to-Q mutant (*Figure 5B*).

## Conformational equilibria influenced by temperature

TM287/288 stems from the hyperthermophilic bacterium *Thermotoga maritima*, which optimally grows at a temperature of 80°C (*Huber et al., 1986*). To investigate the effect of the temperature on the conformational dynamics, we performed several experiments with TM287/288 incubated at 80°C in the presence and absence of nucleotides. Of note, detergent-purified TM287/288 exhibits high ATPase activity at 80°C and therefore is not unfolded or functionally inhibited as a result of heating (*Hohl et al., 2012*). Upon incubation, the samples were snap frozen in cold isopentane to capture a snapshot of the conformational ensemble at physiological temperature. *Figure 6* shows the comparison between distance distributions obtained from samples incubated at 25°C (from *Figure 2*, dotted lines) and 80°C (solid lines).

The biggest temperature-induced changes in the distance distribution were observed in the absence of nucleotides for the spin label pairs placed at the NBDs, in particular for the one reporting changes at the degenerate site (pair 350$^{TM287}$/475$^{TM288}$). Raising the temperature from 25°C to 80°C changed the distance distribution from a monomodal peak centered at 2.5 nm, which is in line with the apo state crystal structure, to a broadly distributed pattern featuring distances from 2 to 8 nm (*Figure 6*). This effect was found to be fully reversible (*Figure 7*) and posits that the degenerate site is largely destabilized at higher temperatures in the absence of nucleotides. A less pronounced effect was observed at the consensus site (pair 460$^{TM287}$/363$^{TM288}$), showing some broadening of the distance distribution, which however was still featuring a prominent peak at 3.5 nm in agreement with the crystallized IF structure (*Figure 6*). The distinct effects observed in the consensus and degenerate sites point to an asymmetric opening of the two nucleotide binding sites at physiological temperature in the absence of nucleotides. DEER measurements on the other intracellular and

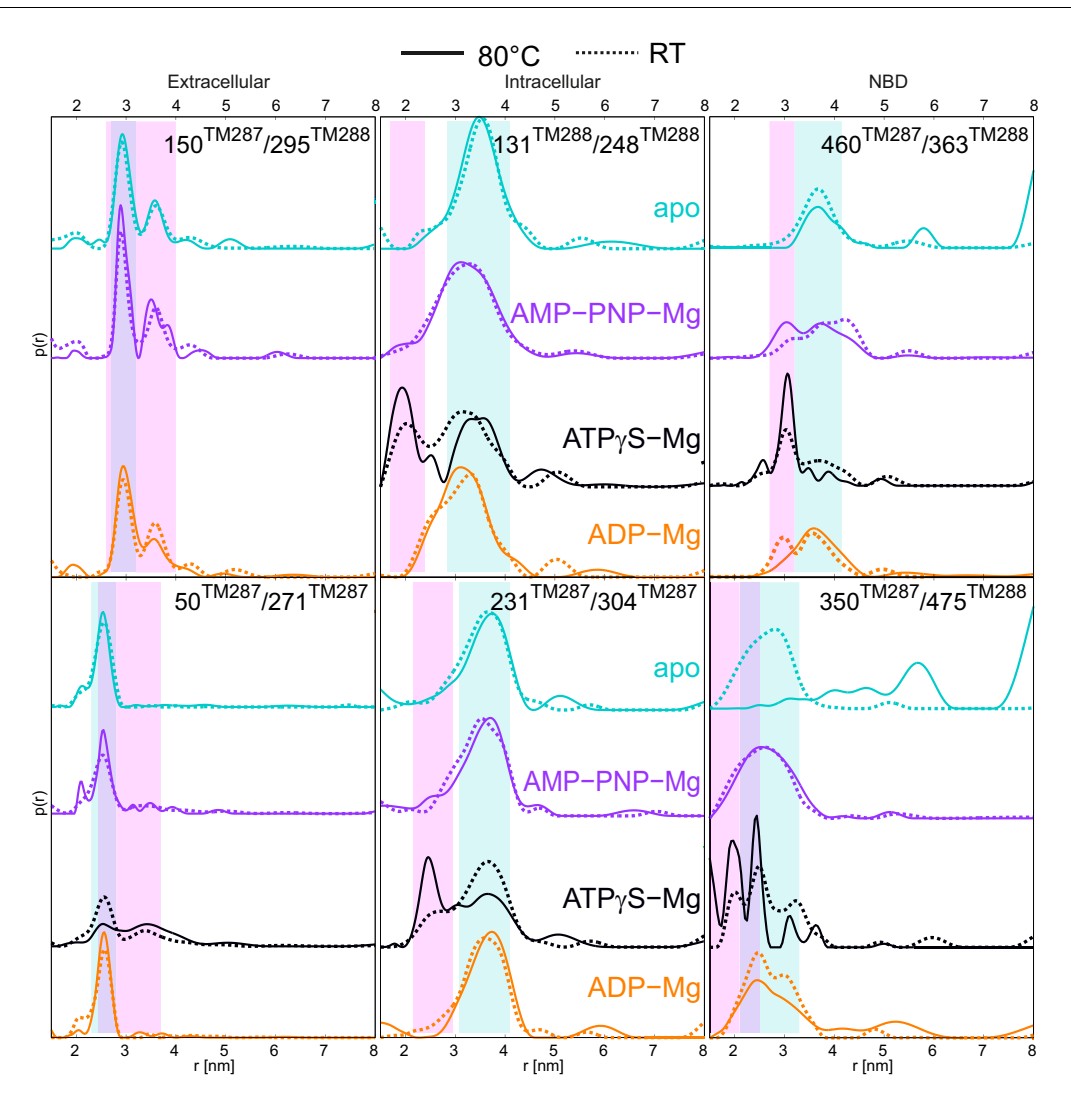

**Figure 6.** DEER distance distribution of samples snap frozen from room temperature and 80°C. Q-band DEER distance distribution for the six spin-labeled pairs engineered on wildtype TM287/288. Distance distributions normalized by area are shown for the samples incubated at room temperature snap frozen in liquid nitrogen (dashed lines, taken from *Figure 2* and *Figure 2—figure supplement 3*) and samples incubated at 80°C and snap frozen in cold isopentane (solid lines). The corresponding DEER data and analysis are shown in *Figure 6—figure supplement 1*.

The following figure supplement is available for figure 6:

**Figure supplement 1.** DEER analysis of wildtype TM287/288 snap frozen from 80°C.

extracellular pairs indicated that the disengagement of the NBDs resulting from incubation at high temperature is not propagated to the TMDs and does not result in a further opening of the inward-facing cavity. In other words, the NBDs are decoupled in terms of dynamics from the rest of the transporter when no nucleotides are present.

Importantly, the presence of any nucleotide (including AMP-PNP-Mg and ADP-Mg) prevented NBD dissociation at 80°C (*Figure 6*). The stabilizing effect of nucleotides on the NBD dimer interface is attributed to the formation of cross-NBD contacts of inward-facing TM287/288. Although AMP-PNP-Mg does not directly bridge the NBD dimer, its presence at the degenerate site leads to a reorientation of several side chains that collectively result in the establishment of a larger number of

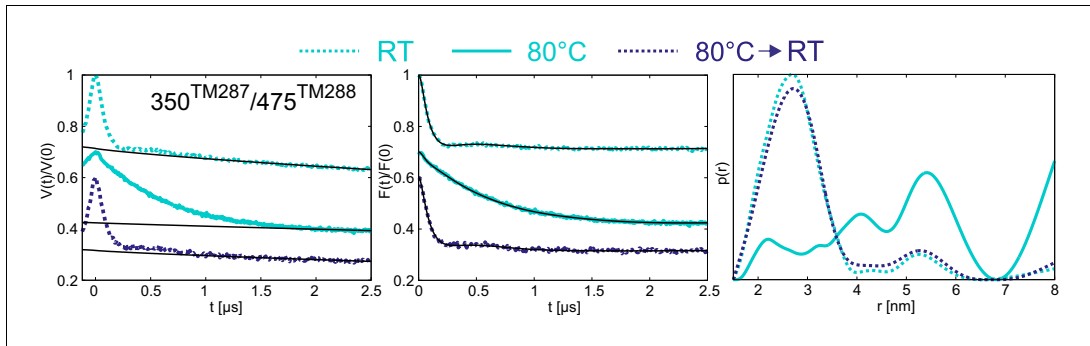

**Figure 7.** Reversibility of the disengagement of the NBDs at high temperatures. Q-band primary DEER data [V(t)/V (0)] with the background fit (left), background corrected Form factor [F(t)/F(0)] with the fit (center) and corresponding distance distribution (right) for the $350^{TM287}/475^{TM288}$ pair monitoring the degenerate site in the absence of nucleotides. Traces were measured using two aliquots of the same sample. The first aliquot was frozen from room temperature (dashed cyan), the second aliquot was frozen from 80°C (solid cyan), then thawed, incubated at room temperature again and snap frozen in liquid nitrogen (dashed blue).

hydrogen bonds across the NBD dimer, mainly mediated by the asymmetric D-loops of the degenerate and the consensus site (*Figure 8*). Similar stabilizing effects are likely to come into play when other nucleotides bind to the NBDs.

DEER measurements conducted with the samples pre-incubated at 80°C in the presence of nucleotide analogs revealed an overall increased propensity to populate the OF state. ATPγS-Mg resulted in a larger fraction of OF states in the distance distribution of all variants with respect to the samples pre-incubated at 25°C. The $K_i$ of ATPγS-Mg decreases with increasing temperature (*Table 2*). However, because ATPγS-Mg was added at saturating concentration, the increased affinity cannot explain the increased population of the OF state. Rather, the temperature dependence of the IF-to-OF transition rate may be steeper than the one of the OF-to-IF transition, resulting in an increased population of the OF state at high temperature. An analogous observation was made for CFTR (*Csanády et al., 2006*). Notably, incubation at 80°C with AMP-PNP-Mg induced a small but detectable fraction of short distances in the distance distribution only in the consensus site (pair $460^{TM287}/363^{TM288}$), analogous to the asymmetric NBD closure observed in the E-to-Q mutant (*Figure 4—figure supplement 2*). However, AMP-PNP does not support the closure of the degenerate site and as a consequence thereof, the transition of the TMDs to the OF state.

Overall, our measurements with spin-labeled TM287/288 incubated at 80°C demonstrate that the transmembrane and extracellular regions of the transporter are only marginally affected by temperature. This is not the case for the NBDs, which partially disengage in an asymmetric fashion in the absence of nucleotides and thereby decouple from the TMDs in terms of dynamics. However, NBD separation in the absence of nucleotides, which we were able to force *in vitro*, is unlikely to occur in the native context of *T. maritima* cells. The cytoplasm contains high concentrations of ATP, and one ATP molecule is bound tightly to the degenerate site of TM287/288 without being hydrolyzed and thereby stabilizes the contacts between the NBDs while the transporter adopts its IF state.

## TM287/288 and BmrCD: two heterodimeric ABC exporters with different energy landscapes

BmrCD is a heterodimeric ABC exporter stemming from the mesophilic bacterium *Bacillus subtilis*. A recent EPR study on BmrCD suggested that this transporter undergoes the conformational transition to the OF state in the presence of ATP-Vi-Mg (i.e. in the vanadate-trapped state), as well as under ATP turnover conditions (i.e. upon addition of ATP-Mg), but not in the presence of non-hydrolyzable AMP-PNP-Mg (*Mishra et al., 2014*). Based on these observations it was concluded that ATP hydrolysis is strictly required for the transition to the OF state, which was called the high-energy post-hydrolysis state (HES) of the transport cycle. A generalized model was put forward which proposed fundamentally different transport mechanisms for homo- and heterodimeric ABC exporters. To verify the results of the BmrCD study and further investigate this transporter with the comprehensive set

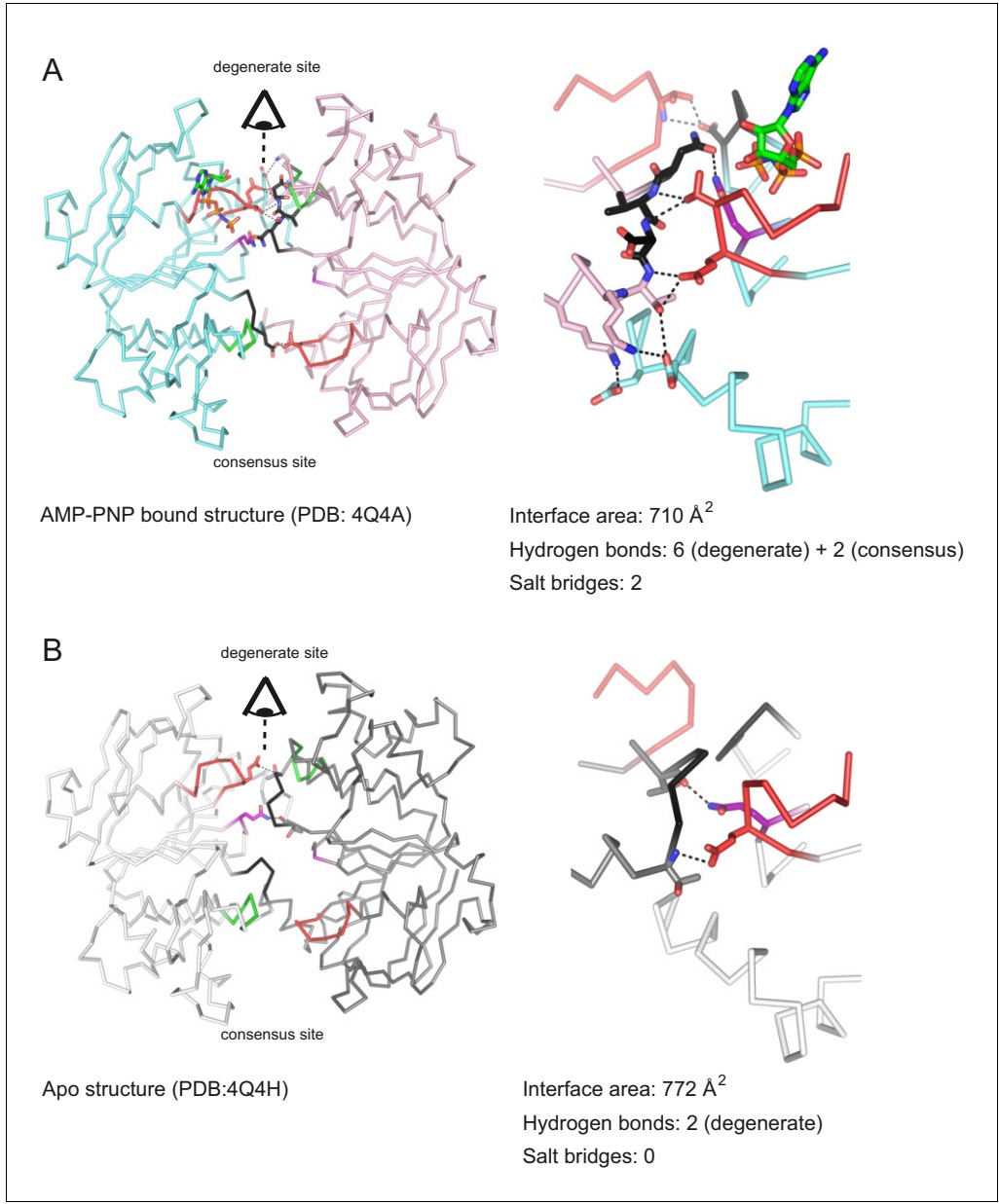

**Figure 8.** Nucleotide binding to the degenerate site stabilizes cross-NBD contacts of inward-facing TM287/288. The interfaces between the NBDs were analyzed by the PISA server. (**A**) Analysis of AMP-PNP-Mg bound TM287/288 (PDB: 4Q4A), (**B**) analysis of apo TM287/288 (PDB: 4Q4A). The left panel shows a top view of the NBDs, the right panel a side view of the degenerate (front) and consensus (back) site. Cross-NBD hydrogen bonds and salt bridges according to the PISA server criteria are highlighted by dashed black lines.

of nucleotides used to study TM287/288 in terms of conformational triggers, we monitored the response of cys-less BmrCD (called wildtype for simplicity) spin labeled at the degenerate site of the NBDs (348[BmrC]/532[BmrD]). In analogy to the E-to-Q mutant of TM287/288, we also introduced the E592Q[BmrD] mutation in the consensus ATP binding site of 348[BmrC]/532[BmrD] (*Table 1*). The response of wildtype BmrCD was found to be fully in accord with the published data (*Mishra et al., 2014*). In the apo state, the distances between the two spin labels are broadly distributed, indicative of a large disorder trapped in the frozen molecular ensemble, which is not further propagated to the TMDs (*Figure 9A*, cyan line). This result is reminiscent of the response of TM287/288 incubated at high temperatures (*Figure 6*). Upon addition of AMP-PNP-Mg, ATPγS-Mg or ADP-Mg a fingerprint

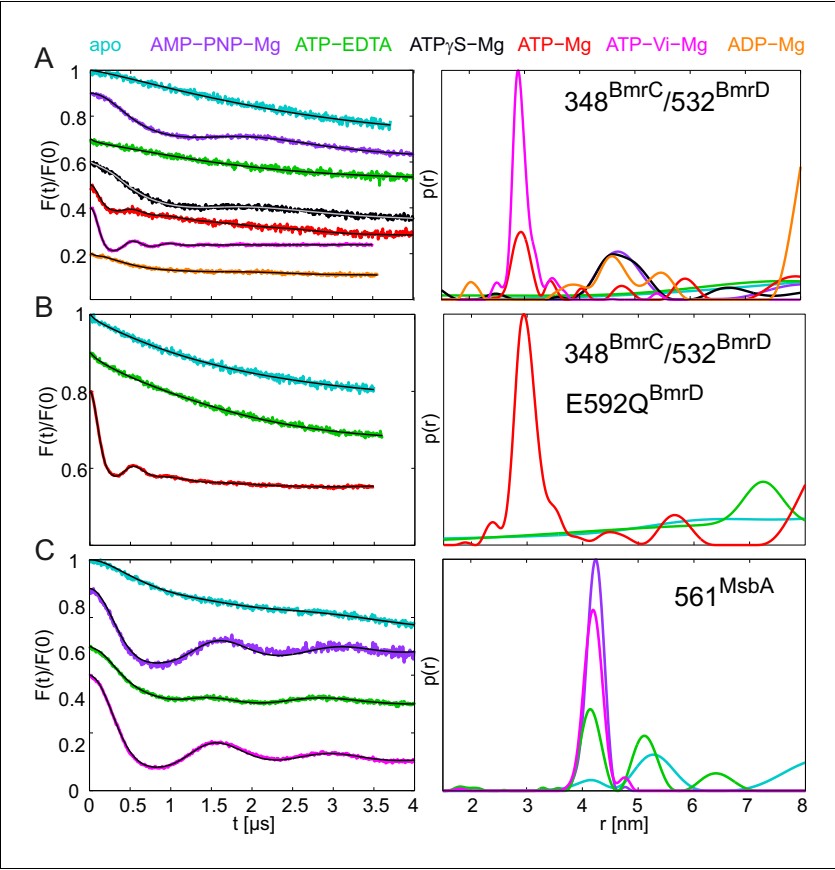

**Figure 9.** DEER analysis of spin-labeled pairs in the NBDs of BmrCD and MsbA. Q-band background-corrected DEER traces [F(t)/F(0)] with fitted distribution function (left) and corresponding distance distribution (right). (**A**) Spin labeled pair $348^{BmrC}/532^{BmrD}$ in wildtype BmrCD and (**B**) in BmrCD carrying the E-to-Q substitution incubated with different nucleotides and nucleotide analogs. (**C**) Spin labeled $561^{MsbA}$ in wildtype MsbA incubated with different nucleotides and nucleotide analogs. Primary DEER traces are shown in *Figure 9—figure supplement 1*. DEER traces detected for BmrCD after incubation with 10 mM nucleotides are shown in *Figure 9—figure supplement 2*.

The following figure supplements are available for figure 9:

**Figure supplement 1.** DEER analysis of BmrCD and MsbA.

**Figure supplement 2.** DEER analysis of wildtype BmrCD in the presence of 2.5 and 10 mM ATP.

distance centered at 4.8 nm appeared in BmrCD, indicative of an IF conformation in which the NBDs are in contact, but not fully closed. This finding is perfectly in line with the IF conformation of TM287/288 under the same experimental conditions (*Figure 9* and *Figure 2—figure supplement 3*). The short distance of 3 nm, which is characteristic for the OF state of BmrCD, was prominently populated upon addition of ATP-Vi-Mg, resulting in vanadate trapping at the consensus site (*Figure 9A*, magenta line). Finally, a quick incubation with ATP-Mg induced the conformational switch to the OF state in a measurable fraction of transporters (*Figure 9A*, red line), i.e. a condition under which the ATPase cycle is operating, which is in agreement with (*Mishra et al., 2014*). In contrast to TM287/288, ATPγS-Mg could only populate the nucleotide-bound state centered at 4.8 nm, but did not result in the switch to the OF state. The $K_i$ for ATPase inhibition of BmrCD at 25°C was determined to be 21.0 ± 3.2 μM and 26.7 ± 3.3 μM for AMP-PNP-Mg and ATPγS-Mg, respectively (*Figure 3E and F*, *Table 2*). Compared to TM287/288, these nucleotide analogs are much weaker inhibitors of ATPase activity. Despite the fact that nucleotides were added at saturating conditions,

poor ATPase inhibition appears to correlate with their inability to close and stabilize the NBD dimer. Addition of 2.5 mM or 10 mM ATP-EDTA did not result in detectable changes of the conformational ensemble with respect to the apo state and in contrast to AMP-PNP-Mg, ATPγS-Mg and ADP-Mg the nucleotide-bound fingerprint distance centered at 4.8 nm was not populated in the absence of $Mg^{2+}$ ions (*Figure 9—figure supplement 2*). Of note, the $Mg^{2+}$ ion is not only required for ATP hydrolysis, but also mediates important contacts between ATP and the Walker A serine. Its omission generally results in a drastic decrease of nucleotide affinity as shown for example for BmrA and TM287/288 (*Hohl et al., 2014*, *Orelle et al., 2003*). ATPase activity measurements revealed that TM287/288 exhibits a very high apparent affinity of 3.27 ± 0.42 μM for ATP-Mg at 25°C, whereas the corresponding affinity amounts only to 343 ± 24 μM for BmrCD (*Table 2*). Due to the lack of ATPase activity in the absence of $Mg^{2+}$, it was not possible to determine a $K_m$ for ATP-EDTA. The $K_m$ for ATP-EDTA may be very high and even the addition of 10 mM ATP-EDTA was probably insufficient to elicit a measurable conformational change as compared to apo BmrCD. Alternatively, ATP-EDTA may have been added at a saturating concentration (as it was the case for TM287/288), but the mere presence of ATP without $Mg^{2+}$ may have been insufficient to mediate NBD contacts as seen in inward-facing TM287/288 or full NBD closure leading to the OF state.

By introducing the E-to-Q mutation in BmrCD, we aimed at monitoring its conformational response in a hydrolytically impaired mutant (*Table 1*). Addition of ATP-EDTA did not change the distance distribution with respect to the apo state, as observed in the wildtype. In contrast, addition of ATP-Mg to the E-to-Q mutant induced a complete switch of the molecular ensemble to the OF state, with the 3 nm fingerprint distance between the two labels (*Figure 9B*). With a turnover rate of 0.09 ATP $min^{-1}$, the residual activity of BmrCD carrying the E-to-Q mutation was found to be higher than the one of TM287/288. During the 15 min of incubation at 25°C, the ATP turnover amounted to 1.35 per transporter, indicating that almost every BmrCD molecule has undergone at least one hydrolysis cycle. Nevertheless, the E-to-Q mutant occluded ATP-Mg in a pre-hydrolytic state at the NBDs, and thereby mainly populated the OF state under these conditions.

## Nucleotide response of the homodimeric MsbA

To extend the analysis of the specific pattern of responses to nucleotides within the ABC exporter family, we monitored the conformational transitions of the homodimeric ABC exporter MsbA using spin labels at position 561 in the NBDs. MsbA has been extensively studied by DEER and it is established that the apo state resembles the inverted V-shaped inward-facing structure with completely separated NBDs. Vanadate trapping results in switching to the OF state, in agreement with the crystal structures (*Borbat et al., 2007*; *Zou et al., 2009*). Here, we show that under the same experimental conditions used for TM287/288 and BmrCD, MsbA partially switched to an OF state upon ATP-EDTA addition (as in TM287/288 but in contrast to BmrCD). Finally, in contrast to both heterodimeric ABC exporters, AMP-PNP-Mg addition supported a complete transition to the OF state in homodimeric MsbA (*Figure 9C*), as shown previously (*Mittal et al., 2012*). Hence, in MsbA containing two consensus ATP binding sites, very similar structural responses are induced by AMP-PNP-Mg and ATP-Vi-Mg (*Figure 9C*). This finding reinforces our notion that AMP-PNP-Mg is incapable of supporting the closure of the degenerate sites of BmrCD and TM287/288.

## Discussion

ABC exporters constitute a very important protein family present in all phyla of life. While recent crystal structures have elucidated the various conformations adopted by ABC exporters in great detail, the nature of the power stroke required for the switch from the IF to the OF state and thereby fueling active substrate transport is a current matter of debate. Here, we studied by EPR how the conformational dynamics of the heterodimeric multidrug transporter TM287/288 is controlled by various nucleotides and a mutation interrupting the ATP hydrolysis reaction. TM287/288 is currently the only heterodimeric ABC exporter featuring asymmetric NBDs with one hydrolysis-incompetent ATP binding site for which a high-resolution structure is available. These reliable atomic models facilitate the DEER analysis and can be directly compared with the data obtained in solution. DEER is a valuable technique, which allows monitoring of the conformational equilibria existing at physiological temperature, by cryo-trapping the molecular ensembles. Our data revealed that in TM287/288 the IF and OF states can be populated simultaneously in the presence of nucleotides.

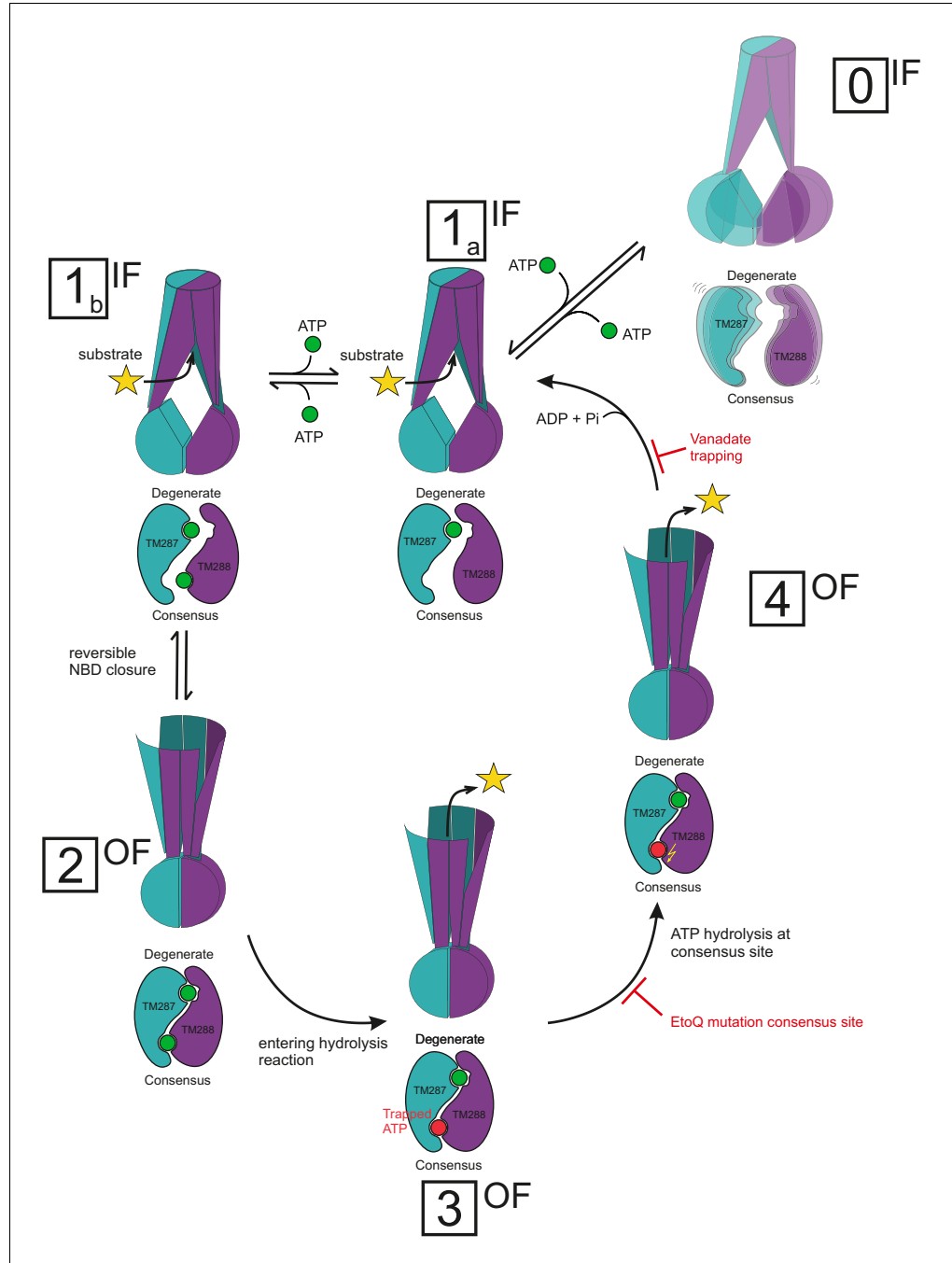

**Figure 10.** Proposed mechanism for heterodimeric ABC exporters. Heterodimeric ABC exporters exhibit the following major conformations: IF with separated NBDs with no nucleotide bound (state 0), IF with partially contacting NBDs and one nucleotide bound to the degenerate site or two nucleotides bound (states 1a and 1b, respectively), OF with closed NBDs and two nucleotides bound (state 2) and stably closed NBDs with a nucleotide trapped at the consensus site in its pre-hydrolysis (state 3) or post-hydrolysis (state 4) state. State 0 is rare and not part of the transport cycle. In state 1, the bound nucleotide does not directly bridge the NBD interface, but causes cross-NBD interactions via the D-loops in an allosteric fashion. Transition from state 1a via state 1b to state 2 is reversible and state 2 is stabilized by nucleotide sandwiching at the NBD interface. Full NBD closure (state 2) is required to initiate ATP hydrolysis at the consensus site. During the hydrolysis reaction, which is an irreversible process, the transporter adopts first a pre-hydrolysis state (state 3), then a post-hydrolysis state (state 4) and upon completion of the ATP hydrolysis reverts to the IF state with one ATP bound at the degenerate site (state 1a). The Walker B E-to-Q mutation in the consensus site and nucleotide trapping by vanadate stabilize the pre- and post-hydrolytic transition state, respectively, and thereby OF states 3 and 4 are populated.

The presence of both conformations proves the existence of an IF-OF equilibrium, which can be modified by the binding of different nucleotides and ATP hydrolysis. Solely ATP binding without hydrolysis – as shown by adding ATP-EDTA to the samples – is sufficient to switch a fraction of transporters from the IF to the OF state. Therefore, our findings call a recent study on BmrCD into question, which claimed that ATP hydrolysis is an absolute requirement for triggering the IF to OF transition in heterodimeric ABC exporters encompassing a degenerate ATP binding site (*Mishra et al., 2014*). Rather, our study highlights the role of a dynamic interplay between coexisting IF and OF states in ABC exporters, which are interconverted in a transporter-specific manner by nucleotide binding and hydrolysis. While the different energy minima populated according to the DEER analyses agree with crystal structures representing IF and OF states, their interconversion is specific to the ABC exporter investigated. For example, MsbA switched completely to the OF state in the presence of AMP-PNP-Mg (*Figure 9C* and [*Mittal et al., 2012*]), TM287/288 and BmrCD were instead stabilized in an IF state in which the NBDs were in close contact (*Figure 2* and *Figure 9*). Interestingly, AMP-PNP was also shown to prevent the switching to the OF state in SUR1, the regulatory subunit of the neuroendocrine ATP-sensitive K$^+$ channel (*Ortiz et al., 2013*) and it is known to be a poor nucleotide analog to switch CFTR to its chloride conducting OF state (*Vergani et al., 2003*). Another example is the different response among ABC exporters with regard to ATP-EDTA: LmrA completely switched to the OF state (*Hellmich et al., 2012*), BmrCD did not (*Figure 9A*) and MsbA and TM287/288 did to some extent (*Figure 2* and *Figure 9*).

The molecular events discovered allowed us to build a mechanistic model, which reconciles all DEER data available for heterodimeric ABC exporters (*Figure 10*), but contradicts the recently proposed mechanism for heterodimeric ABC exporters (*Mishra et al., 2014*). It is undisputed that ATP binding and hydrolysis at the NBDs drive active transport by ABC exporters and – with a few exceptions – the field agrees that a typical ABC exporter adopts two principal states, namely an IF state with NBDs fully or partially separated and an OF state with nucleotides bound at the closed NBD dimer interface. The debated key questions, however, deal with the molecular events leading from an open to a closed NBD dimer having two nucleotides sandwiched at the interface, thereby acting as molecular glue, and with the coupling of these movements to the TMDs.

All ABC exporters studied by DEER are strongly shifted towards the IF state in the absence of nucleotides. In this study, we confirm this notion for MsbA and BmrCD and show that in TM287/288 at physiologically high temperatures, the IF conformation features separated NBDs, which are decoupled from the rest of the transporter, as in BmrCD. This decoupling unraveled a level of dynamic freedom between the engine and the TMDs which was not expected based on the coupling through the two main and two accessory coupling helices observed in crystallized states. Of note, in the living cell, ATP levels are (in average) in the millimolar range (*Yaginuma et al., 2014*), so that the apo state is a very rare event, namely when nucleotides are exchanged at the same time at both ATP binding sites. This is particularly true in the case of heterodimeric ABC exporters, which firmly hold an ATP in the high affinity degenerate site without hydrolyzing it. Therefore, we consider the apo state with the engine decoupled from the rest of the machine as an 'off-state' (state 0 in *Figure 10*), which is rare and not required in the transport cycle. Nevertheless, it is worth noting that the ATP levels greatly differ at the single cell level with some cells being devoid of detectable ATP concentrations (*Yaginuma et al., 2014*). In such ATP-deprived cells, the functional relevance of the decoupling between NBDs and TMDs remains to be elucidated.

Heterodimeric ABC exporters such as TM287/288 and BmrCD feature a degenerate ATP binding site, which binds nucleotides tightly without hydrolyzing them. In the crystal structure of TM287/288 AMP-PNP was shown to bind exclusively to the degenerate site. In contrast to ABC exporters with two consensus sites crystallized in IF states, which exhibit fully separated NBDs, the NBDs of inward-facing TM287/288 are in molecular contact (state 1a in *Figure 10*). Importantly, AMP-PNP does not directly mediate cross-NBD contacts as those found in closed NBDs (*Figure 8*). Instead, a comparison of the apo and the AMP-PNP-Mg bound structures of TM287/288 revealed that nucleotide binding at the degenerate site results in the formation of six additional hydrogen bonds and two novel salt bridges between the NBDs via long-range conformational changes mainly involving the two asymmetric D-loops (*Figure 8*). The functional importance of nucleotide binding to the degenerate site resulting in a NBD-NBD contact in the inward-facing transporter can be appreciated in both heterodimeric ABC exporters studied so far. In the case of BmrCD in the absence of nucleotides, the NBDs are fully separated, akin to structures of inward-facing ABCB1 or MsbA. However, in the

presence of the nucleotides AMP-PNP-Mg, ATPγS-Mg and ADP-Mg, the NBDs get closer, with a fingerprint distance, which agrees with NBD-NBD contact as seen in the TM287/288 IF structures. In case of TM287/288, heating to 80℃ in the absence of nucleotides was required to obtain an IF state with asymmetrically separated NBDs. Again, the presence of AMP-PNP-Mg, ATPγS-Mg and ADP-Mg prevented NBD separation at 80℃, maintaining the transporter in an IF state resembling the crystal structures. Hence, as rationalized by the comparison of AMP-PNP-Mg bound and apo TM287/288 structures, nucleotide binding to the degenerate site (and possibly also to the consensus site as shown in state 1b), even including the hydrolysis product ADP, prevents NBD separation under physiological conditions in TM287/288 and BmrCD (*Figure 10*). This distinctive structural feature differentiates the heterodimeric ABC exporters BmrCD and TM287/288 from the homodimeric ABC exporters studied by DEER, namely MsbA and LmrA, which were shown to cycle only between two states characterized by fully separated and fully closed NBDs.

Importantly, the partial NBD contacts of inward-facing TM287/288 are not mediated directly by nucleotide sandwiching, but rather via the D-loops (*Figure 8*). These contacts facilitate NBD closure, because the NBDs remain appropriately aligned. By contrast, fully separated NBDs of homodimeric ABC exporters have to sample a much larger conformational space for their correct dimerization. Nucleotides only act as molecular glue, thus creating the closed NBD sandwiched dimer, at the end of the IF to OF state trajectory by bridging the Walker A motif of the first with the ABC signature motif of the second NBD and *vice versa* (state 2 in *Figure 10*). In other words, nucleotide binding to fully or partially separated NBDs of inward-facing ABC exporters is unlikely a requirement for NBD closure. Instead, the transition between opened and closed NBDs – and consequently – the transition between the IF and OF state likely occurs spontaneously by Brownian motion even in the absence of nucleotides. This was demonstrated previously by cross-linking experiments on TM287/288 (*Hohl et al., 2012*), cryo-EM studies on MsbA (*Moeller et al., 2015*) and for CFTR, which opens its pore spontaneously at low frequency even in the absence of nucleotides (*Mihályi et al., 2016*). To stabilize the closed NBD dimer by two sandwiched nucleotides resulting in an OF orientation of the TMDs (state 2 in *Figure 10*), a second nucleotide has to bind to the consensus site (state 1b) during the process of NBD closure. Our model envisages that the interconversion between states 1b and 2 is reversible as long as the consensus site nucleotide is not committed to hydrolysis. This was demonstrated by adding ATP-EDTA at saturating concentration, which resulted in the population of a measurable fraction of OF state as a result of mere nucleotide binding. However, this was not the case for BmrCD, where a very low ATP-EDTA affinity or a strict requirement of $Mg^{2+}$ ions likely prevented the formation of a detectable population of the OF state. Importantly, hydrolysis can only occur after NBD closure, hence the IF to OF transition must be driven by nucleotide binding rather than hydrolysis. Once the ATP at the consensus site commits to the hydrolysis reaction, the transporter irreversibly progresses first to a pre-hydrolytic and then to a post-hydrolytic state (states 3 and 4 in *Figure 10*), which are structurally indistinguishable from state 2 by DEER with the available spin-labeled pairs. The molecular event of entering the hydrolysis reaction – also called nucleotide occlusion in earlier studies on ABCB1 (*Sauna and Ambudkar, 2001*; *Sauna et al., 2006*, *2007*) – strongly increases nucleotide affinity (highlighted by a red colored ATP in state 3, *Figure 10*) and thereby stabilizes the closed NBD sandwich dimer. The coupling between NBDs and TMDs ensures an efficient propagation of the structural rearrangements, which leads to a prolonged OF configuration of the transporter and permits substrate release (states 3 and 4 in *Figure 10*). In the active transporter, ATP is hydrolyzed to ADP and $P_i$, which reverts the transporter back to its IF state with a nucleotide still bound to the degenerate binding site (irreversible transition from state 4 to state 1 in *Figure 10*). Progression of the hydrolysis reaction is slowed down by stabilization of the pre- or the post-hydrolytic state. In our study, we stabilized the pre-hydrolytic state by mutating the catalytically important Walker B glutamate of the consensus site into a glutamine (E-to-Q mutation). Further, we stabilized the post-hydrolytic state by vanadate trapping. In both regimes, a bound ATP commits to the hydrolysis reaction at the consensus site, but the hydrolysis rate is drastically slowed down, so that the transporters are trapped in the OF state (states 3 and 4 in *Figure 10*). In essence, entering the hydrolysis reaction appears to play an essential role in stabilizing the closed NBD dimer without being an absolute requirement to support the IF to OF switch in heterodimeric or homodimeric ABC exporters.

The E-to-Q mutation at the consensus site slows down the hydrolysis reaction of a trapped nucleotide resulting in an increased population of the OF state upon binding of hydrolyzable nucleotides

such as ATP-Mg and ATPγS-Mg. Surprisingly, similar fractions of the OF state could be populated in the E-to-Q mutant with ATP-EDTA under strict non-hydrolytic conditions. An analogous observation was reported for SUR1, which was attributed to an increased affinity of ATP-EDTA for the transporter as a consequence of the removal of the negatively charged catalytic glutamate (*Ortiz et al., 2013*). However, the IF-OF ratio observed in TM287/288 incubated with 2.5 and 14 mM ATP concentrations in the absence of $Mg^{2+}$ ions was found to be almost indistinguishable. This indicates that 2.5 mM ATP already represented a saturating concentration for the wildtype, and consequently, for the E-to-Q mutant as well. Therefore, our data suggest that the E-to-Q mutation modulated the energy profile of TM287/288 by shifting the equilibrium towards the OF state.

In agreement with this notion, a partial closure at the NBD's consensus site of the E-to-Q mutant was induced by addition of saturating concentrations of the non-cleavable nucleotide analog AMP-PNP-Mg. Not only the E-to-Q mutant, but also a temperature increase from 25°C to 80°C shifted the equilibrium to the OF state, and permitted the detection of asymmetric NBD closure in the presence of AMP-PNP-Mg. Interestingly, AMP-PNP-Mg turned out to be unable to close the degenerate site. This surprising finding explains why AMP-PNP-Mg firmly closes the NBDs of homodimeric MsbA by binding to the two consensus sites but not those of the heterodimeric ABC exporters TM287/288 and BmrCD.

In metabolically active cells, wildtype ABC exporters constantly hydrolyze ATP (basal ATPase). We mimicked this situation *in vitro* by incubating TM287/288 with ATP-Mg for a short time and observed an approximate 1:1 mixture of IF and OF states. Similar fractions between IF and OF states during ATP hydrolysis were observed with LmrA, MsbA and BmrCD (*Mishra et al., 2014*; *Hellmich et al., 2012*). ABC exporters have therefore evolved an energy profile that permits them to adopt the functionally relevant conformational states under operating conditions. The inability of the catalytically impaired E-to-Q mutant to transport substrates can be explained by an overpopulation of the OF state, which prevents the transporter from switching forth and back between its IF and OF states.

Recent crystallographic studies revealed an outward-occluded state of ABC exporters, in which the NBDs are fully closed, but the extracellular gate is not (yet) opened (*Choudhury et al., 2014*). Our DEER measurement with two spin-labeled pairs in TM287/288 located in the extracellular region revealed that under conditions in which the NBDs are fully closed, this region is characterized by a broad distance distribution encompassing distances already present in the apo-state when the NBDs are disengaged. This indicates that the extracellular gate of TM287/288 is a highly dynamic region of the transporter. Based on our data we cannot rule out that the extracellular gate possibly covers a conformational ensemble between outward-occluded and outward-facing states. A recent biophysical study of CFTR elucidated a conformational progression of channel opening with the following sequence of events: NBD closure, propagation of the conformational change to the intracellular part of the TMDs and finally the opening of the extracellular gate (*Sorum et al., 2015*). The functional importance of extracellular gate opening in the context of substrate transport remains elusive and studying the effects of substrates or point mutations in the transporter will help to elucidate this intriguing aspect.

## Conclusions

Each ABC exporter studied thus far by DEER appears to possess an inbuilt, characteristic energy landscape, which has evolved to pump its dedicated substrates in the presence of high physiological ATP concentrations. Although BmrCD and TM287/288 are both heterodimeric ABC exporters, they behave very differently with regard to their conformational dynamics. In BmrCD, the two IF states (with partial NBD contact or fully separated NBDs) are clearly preferred over the OF state, while from a thermodynamic point of view the two major IF and OF states populated in the presence of nucleotides are closer together in TM287/288. This explains the differences seen between the two heterodimeric ABC exporters; while the conformational switch of BmrCD could only be achieved by ATP-Mg entering a hydrolysis reaction (pre-hydrolytic by E-to-Q mutation or post-hydrolytic by vanadate trapping), nucleotide binding without ATP hydrolysis was sufficient for the switch in TM287/288. Our results therefore do not support the notion that heterodimeric ABC exporters strictly require ATP hydrolysis to switch to the OF state.

The heterodimeric ABC exporters BmrCD and TM287/288 dispose of a hydrolysis-incompetent, degenerate ATP binding site, whose 'raison d'être' remains to be fully revealed. DEER

measurements presented in this work together with previously obtained crystals structures of TM287/288 show that nucleotide binding to the degenerate site strongly stabilizes cross-NBD interactions mediated by the asymmetric D-loops while the TMDs adopt an IF state. Hence, an important functional role of the degenerate site is to prevent full separation of the NBDs as observed in homodimeric ABC exporters. Consequently, this pre-orientation facilitates NBD closure and may be relevant for substrate binding and regulatory mechanisms in these ABC exporters.

Energy landscapes likely have profound effects on the function of each ABC exporter. The energy profile governs the dwelling time of the transporter in the various states of the transport cycle, thereby kinetically tailoring the interaction with substrates and regulatory partners on either side of the membrane and the ATP hydrolysis reaction at the NBDs. An altered energy profile may be the underlying mechanism behind many unexpected phenotypes of transporter point mutants. A noteworthy example to be studied in this respect is a mutation in the NBDs of the multidrug transporter Pdr5, which changed the drug transport profile of this transporter (*Ernst et al., 2008*). Future studies of thermodynamic and kinetic parameters of ABC transporters will likely reveal unprecedented clues about their function and molecular mechanism.

## Materials and methods

### Selection of spin-labeled pairs and interspin distance simulations

Optimal spin-labeled pair positions were selected using the available PDB data and the rotamer library approach for the MTSL ([1-Oxyl-2,2,5,5-tetramethyl-Δ3-pyrroline-3-methyl] Methanethiosulfonate) spin-labeled side chains available in the Matlab package MMM2015 (*Polyhach et al., 2011*). Two following rotamer libraries were used: the MTSL library at 175 K available in the MMM2013 version, and the new library at ambient temperature from the MMM2015 version. The following criteria were used: strategic position in the whole transporter, high accessibility towards the aqueous environment, high number of populated rotamers and distinct distances in the inward with respect to the outward-facing models. For the in-silico analysis, the inward-facing crystal structure of TM287/288 in its apo state (PDB: 4Q4H) and the outward-facing structure created by homology modeling using Sav1866 (PDB: 2HYD) were used as templates.

### Protein preparation and activity assay

The expression vectors containing cys-less versions of TM287/288 (called wildtype TM287/288 for simplicity) (*Hohl et al., 2012*) and TM287/288_ E517Q$^{TM288}$ were used as templates to generate the spin labeling mutants. BmrCD was cloned from chromosomal DNA of *B. subtilis* into the expression vector pBXNH3 using FX-cloning (*Geertsma and Dutzler, 2011*). The three cysteines of BmrD were replaced by alanines by site-directed mutagenesis. The expression vectors encoding cys-less BmrCD (called wildtype BmrCD for simplicity) and BmrCD_E592Q$^{BmrD}$ served as templates to generate the DEER mutant 348$^{BmrC}$/532$^{BmrD}$, which places spin labels at the degenerate site and has been studied in detail before (*Mishra et al., 2014*). Cys-less MsbA cloned into pBAD24 (*Mittal et al., 2012*) was used as template to introduce the spin-labeled pair at position 561. TM287/288, BmrCD and MsbA were expressed in *E. coli* MC1061, purified and spin labeled as described elsewhere (*Hohl et al., 2012, 2014*; *Mittal et al., 2012*). Purification was carried out in the presence of 2 mM DTT with β-DDM as detergent for TM287/288 and BmrCD. MsbA was prepared identically, but β-UDM was used as a detergent. ATPase activity measurements with detergent-purified protein were carried out in ATPase buffer consisting of 20 mM Tris pH 7.4, 150 mM NaCl, 10 mM MgSO$_4$ containing 0.03% β-DDM (for TM287/288 and BmrCD) or 0.05% β-UDM (for MsbA). Liberated phosphate was detected using molybdate/malachite green detection as described (*Hohl et al., 2014*). To determine K$_m$ values, ATPase assay was carried out at varying ATP concentrations. To determine the IC$_{50}$ for ADP-Mg, AMP-PNP-Mg, ATPγS-Mg and vanadate, wildtype TM287/288 and TM287/288_E517Q$^{TM288}$ (for background subtraction) were incubated in ATPase buffer with 500 μM ATP for 30 min at 25°C or for 10 min at 50°C with increasing concentrations of the nucleotide analogs or vanadate. In case of BmrCD, wildtype and BmrCD_E592Q$^{BmrD}$ (for background subtraction) were incubated in ATPase buffer containing 2.5 mM ATP for 30 min at 25°C. In order to minimize detection problems caused by free phosphate of the competitor nucleotides (in particular AMP-PNP, Order No. A2647, Sigma-Aldrich), the reaction volume was reduced from 90 μl to 10 μl in the reactions

used for $IC_{50}$ determination. The data were fitted to a hyperbolic decay curve with the following function (SigmaPlot):

$$f = y0 + (a \cdot IC_{50})/(IC_{50} + x),$$

in which $f$ corresponds to the ATPase activity at the respective inhibitor concentration divided by the ATPase activity in the absence of inhibitor normalized to 100%, $y0$ corresponds to the residual activity at infinite inhibitor concentration, $a$ corresponds to the maximal degree of inhibition ($a + y0 = 100\%$) and $x$ corresponds to the inhibitor concentration.

$K_m$ values reporting the apparent ATP affinity in the absence of inhibitor were determined by measuring the ATPase activity at varying ATP concentration and the curves were fitted with the Michaelis Menten equation:

$$f = v_{max} \cdot x/(K_m + x),$$

in which $f$ corresponds to the ATPase activity and $x$ corresponds to the ATP concentration.

$K_i$ values were calculated according to the formula

$$K_i = IC_{50}/(1 + [S]/K_m)$$

in which $[S]$ corresponds to the ATP concentration used for the determination of the respective $IC_{50}$ values. $IC_{50}$ and $K_m$ values were obtained from data fitting as outlined above.

Standard errors for the $K_i$ values were calculated by error propagation:

$$\sigma_{Ki} = \sqrt{\left(\frac{K_m}{K_m + [S]}\right)^2 \sigma_{IC_{50}}^2 + \left(\frac{[S]IC_{50}}{(K_m + [S])^2}\right)^2 \sigma_{K_m}^2}$$

in which the $\sigma_{IC50}$ and $\sigma_{Km}$ are standard errors obtained from data fitting in SigmaPlot and $\sigma_{Ki}$ is the standard error for $K_i$.

In order to determine residual ATPase activities of TM287/288_E517Q$^{TM288}$ and BmrCD_E592Q$^{BmrD}$, 3.4 µM or 1 µM of the respective transporter was incubated at 25°C in ATPase buffer with 1 mM and 2.5 mM ATP, respectively. Incubation with buffers containing the same nucleotide concentration without protein served for background subtraction in these assays. Residual TM287/288-mediated nucleotide hydrolysis in the presence of 0.5 mM ATP/2.5 mM EDTA or 0.25 mM AMP-PNP-Mg was assessed by incubating 2 µM of the purified wildtype transporter for 30 min at 50°C.

To carry out DEER measurements in a lipidic environment (pair 131$^{TM288}$/248$^{TM288}$) as well as for the investigation of drug-stimulated ATPase activity of all pairs, spin-labeled TM287/288 was reconstituted into proteoliposomes consisting of *E. coli* polar lipids and egg phosphatidylcholine mixed at a ratio of 3:1 in 50 mM K-HEPES pH 7.0 as described elsewhere (*Hürlimann et al., 2016*; *Geertsma et al., 2008*). To determine stimulation of ATPase by Hoechst 33342, proteoliposomes containing reconstituted wildtype and spin-labeled TM287/288 was pre-incubated by 0 µM, 50 µM, 100 µM and 150 µM Hoechst 33342 and ATPase activity was determined in 50 mM K-HEPES pH 7.0, 10 mM MgSO$_4$ in the presence of 1 mM ATP. Reconstituted TM287/288_E517Q$^{TM288}$ was used for background subtraction.

## EPR sample preparation

The labeling efficiency of the double cysteine mutants of the transporters solubilized in detergent was measured at 25°C using an X-band Miniscope 400 EPR spectrometer (Magnettech by Freiberg Instrument). Samples were loaded into glass capillaries with the inner diameter of 0.9 mm and spectra were measured with 14 mT field sweep, 0.15 mT modulation amplitude, 2.5 mW microwave power. The calculated spin labeling efficiencies of the twelve mutants ranged between 70–90% (the second integral of the EPR spectra was calculated with the software available at spintoolbox.com). The high degree of labeling is correlated with the 0.3–0.4 modulation depths of all Q-band DEER traces presented. For DEER measurements, 10% v/v D$_8$- glycerol was added prior to snap freezing. The range of final protein concentrations was 12 to 41 µM. 40 µL of sample were loaded in quartz tubes with 3 mm outer diameter. If not stated differently, samples were incubated at 25°C for 10 min (TM287/288 and MsbA) or 15 min (BmrCD) and snap frozen in liquid nitrogen. To study the

effect of high physiological temperatures on the frozen dynamics of the transporter, samples were loaded in the quartz tubes, incubated for 3 min at 80°C on a heat block and snap frozen in a mixture of isopentane and liquid nitrogen to minimize the freezing time. The ATP-Mg samples were incubated with 2.5 mM ATP and 2.5 mM $MgCl_2$ for the E-to-Q mutant. In the case of wildtype TM287/288, ATP, $MgCl_2$ and 10% v/v $D_8$-glycerol were added and the samples were snap frozen within 20 s to minimize ATP turnover. To block ATP hydrolysis, 2.5 mM EDTA was added prior addition of 2.5 mM ATP. In certain measurements, 10 mM or 14 mM ATP was added (in the presence of 2.5 mM EDTA) for BmrCD and TM287/288, respectively, as indicated in the corresponding figure legends. The nucleotide analogs AMP-PNP-Mg and ATPγS-Mg (2.5 mM) were also prepared in the presence of 2.5 mM $MgCl_2$. In case of ADP-Mg, 2.5 mM ADP and 2.5 mM $MgCl_2$ were added. For vanadate trapping, samples were incubated with 5 mM sodium orthovanadate, 2.5 mM ATP and 2.5 mM $MgCl_2$ in the presence of 10% v/v $D_8$-glycerol for 3 min at 50°C for TM287/288 (25°C for BmrCD) and snap frozen in liquid nitrogen.

### DEER analysis

Double electron-electron resonance (DEER) measurements were performed at 50 K on a Bruker ELEXSYS E580Q-AWG (arbitrary wave generator) dedicated pulse Q-band spectrometer equipped with a 150 W TWT amplifier. A 4-pulse DEER sequence with rectangular, non-selective observer and pump pulses of 14 or 16 ns length (depending on the available $B_1$ at the sample) with 100 MHz frequency separation was used (*Polyhach et al., 2012*). Due to the coherent nature of the AWG generated pulses, a four-step phase cycling $(0 – \pi/2 - \pi – 3/2\pi)$ of the pump $\pi$ pulse was performed to remove unwanted effects of running echoes from the DEER trace. The evaluation of the DEER data was performed using DeerAnalysis2015 (*Jeschke et al., 2006*). The background of the primary DEER traces was corrected using stretched exponential functions with homogeneous dimensions of 1.5 to 3 for different samples. A model-free Tikhonov regularization was used to extract distance distributions from the background corrected form factors. The data of the apo and AMP-PNP-Mg states of the four pairs $150^{TM287}/295^{TM288}$, $131^{TM288}/248^{TM288}$, $460^{TM287}/363^{TM288}$, $350^{TM287}/475^{TM288}$ in the wildtype transporter are taken from (*Hohl et al., 2014*).

To extract the fraction of OF and IF states in the distribution, a model-based two-Gaussian fitting was also used, which generally resulted in a lower quality of the fit of the form factors. The fit was first performed on the ATP-Vi-Mg state, and the obtained mean distance and width (σ) of the major OF distribution were found, and kept fixed for the fit of all other states. The parameters of the second Gaussian were allowed to vary to take into account small variation in the distributions in different states. More than two protein batches were used for all mutants investigated and the DEER data at room and high temperature were found to be highly reproducible.

## Acknowledgements

EB would like to thank G Jeschke (ETH Zurich) for providing the Q-band resonator and R Schlesinger (FU Berlin) for providing access to the molecular biology lab. MAS thanks the Institute of Medical Microbiology and the University of Zurich for financial and administrative support. EB acknowledges support by the Deutsche Forschungsgemeinschaft (DFG) through grants BO3000/1–2 and INST 130/972–1 FUGG. This work is supported by the DFG Cluster of Excellence RESOLV (EXC 1069, to EB) funded by the Deutsche Forschungsgemeinschaft and by a SNF Professorship of the Swiss National Science Foundation (PP00P3_144823, to MAS).

## Additional information

### Funding

| Funder | Grant reference number | Author |
|---|---|---|
| Schweizerischer Nationalfonds zur Förderung der Wissenschaftlichen Forschung | PP00P3_144823 | Markus A Seeger |
| Deutsche Forschungsgemeinschaft | Cluster of Excellence RESOLV EXC 1069 | Enrica Bordignon |

| Deutsche Forschungsge-meinschaft | INST 130/972-1 FUGG | Enrica Bordignon |
| Deutsche Forschungsge-meinschaft | BO3000/1-2 | Enrica Bordignon |

The funders had no role in study design, data collection and interpretation, or the decision to submit the work for publication.

## Author contributions

MHT, CAJH, SB, Acquisition of data, Analysis and interpretation of data, Drafting or revising the article; MH, TA, AM, Acquisition of data, Drafting or revising the article; MAS, EB, Conception and design, Acquisition of data, Analysis and interpretation of data, Drafting or revising the article

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
