## [Decision Letter]

Thank you for submitting your article "Exploring conformational equilibria of a heterodimeric ABC transporter" for consideration by *eLife*. Your article has been reviewed by two peer reviewers, and the evaluation has been overseen by Kenton Swartz as the Reviewing Editor and Richard Aldrich as the Senior Editor. The following individual involved in review of your submission has agreed to reveal his identity: László Csanády (Reviewer #1).

The reviewers have discussed the reviews with one another and the Reviewing Editor has drafted this decision to help you prepare a revised submission.

Summary:

This impressive study uses Double Electron Electron Resonance (DEER) to measure state-dependent changes in distances between pairs of amino acid residues in a heterodimeric ABC exporter, TM287/288. A high-resolution crystal structure solved previously by the authors was used to define suitable pairs that track distance changes between opposing nucleotide binding domain (NBD) surfaces both in the canonical and the degenerate nucleotide binding site, and between the intracellular or extracellular ends of pairs of transmembrane α helices. These measurements enable the authors to quantify the steady-state (under hydrolytic conditions) or equilibrium (under non-hydrolytic conditions) distribution of the transporter between inward-facing (IF) and outward-facing (OF) transmembrane domain conformations, and between open (disengaged) and closed (dimerized) conformations of either of the two interfacial nucleotide binding sites. Using this powerful approach the authors demonstrate that (i) the IF conformation is coupled with an open canonical and a partially open degenerate nucleotide binding site, while the OF conformation is coupled with dimerized NBDs, (ii) different nucleotides are differentially efficient in shifting the IF<->OF conformational distribution, (iii) nucleotide hydrolysis is not required for the IF->OF step, (iv) the distribution can be shifted towards the OF conformation by trapping the transporter in a prehydrolytic (E-to-Q mutation of the catalytic base) or posthydrolytic (using ATP+vanadate) state (v) or by raising temperature, and (vi) that distinct differences in conformational energetics can be detected for different asymmetric and symmetric ABC exporters.

These results provide new insight into the conformational dynamics of heterodimeric ABC exporters, and beautifully demonstrate how the DEER technique can be used to track conformational movements in proteins. The experiments seem to be implemented carefully, and are clearly described. However, we do feel that some aspects of the interpretation, and of the mechanistic conclusions drawn, are internally inconsistent, and would benefit from some deeper re-thinking. Below are suggestions, based on the data presented here and on earlier work on other ABC proteins, for improving those aspects of the presentation.

Essential revisions:

1) We feel that ATPase activity measurements for the 6 pairs of labeled Cys residues is not sufficient to conclude the transporter is functional. We request that the authors provide measurements of transport or at least substrate-stimulated ATPase activity for all six pairs of spin-labeled Cys mutants.

2) DEER measurements in the presence of Mg-ATP for the 150/295 pair is conspicuously missing from the figures. Could the authors provide this data to complete the dataset?

3) It is not clear how the range of distance characteristics of the IF and OF conformation as given in Figure 1,Figure 4,Figure 6 and the supplements were achieved. In some cases (e.g. 150/295, 460/363) they clearly deviate from the predicted distances using MMM. Nonetheless, the authors stated that the distances for at least the IF conformation agree with the crystal structure. This is not always true as can be seen in Figure 1—figure supplement 1 and should be clarified and discussed in light of possible alternative conformations.

4) Competition studies using ATP hydrolysis assay (Figure 3)

4.1) Fitting such dose response curves (fractional ATP hydrolysis turnover rates at a fixed [ATP], plotted as a function of competitor concentration) with sums of exponentials is entirely inadequate: no mechanistic model predicts exponential dose response relationships. Instead these curves should be fitted with hyperbolic functions of the form v/v0=IC50/(<I>+IC50), where <I> is the concentration of the competitor, and v0 is the rate measured at <I>=0. Using logarithmic abscissae might also help visualizing these IC50 values.

4.2. The IC50 values themselves are not very informative since they depend on the concentration of the substrate (ATP) as follows: IC50=K_i_(1+[S]/K_m_), where K_i_ is the apparent affinity of the inhibitor, [S] is the concentration of the substrate, and K_m_ is the Michaelis constant for ATP hydrolysis in the absence of an inhibitor. (The above equations assume competition for a single site, but such a simplification does not seem inappropriate, since at the very high [ATP] used the high-affinity degenerate site likely remains loaded with ATP.) Thus, given a K_m_ of ~20 μm for ATP (line 376), and [ATP]=500 μm in these assays, the relationship IC50=~26xK_i_ holds. The IC50 values of 44 μm, >800 μm, and 565 μm for ATPgS, AMP-PNP, and ADP (line 200) therefore correspond to K_i_values (binding affinities) of 1.7 μm, >31 μm, and 22 μm, respectively. In addition to the IC50 values, these calculated K_i_values should also be provided for each nucleotide, transporter, and temperature, as these are highly relevant for the interpretation of the DEER experiments performed in the presence of those nucleotides alone (see comment 5., below).

5) Interpretation of nucleotide efficiencies to stabilize OF state

Efficiencies of various nucleotides to stabilize the OF state are repeatedly interpreted to reflect the affinities of the respective nucleotides (subsection “Conformational equilibria explored in ATPase deficient E-to-Q mutant of TM287/288”, subsection “Conformational equilibria influenced by temperature”, fourth paragraph and Discussion, fifth paragraph). However, this interpretation is incorrect, since – except for the apo condition – the DEER measurements were all performed at saturating nucleotide concentrations (2.5 mM, whereas K_i_values are low μM for each; see comment 4., above). Instead, the fraction of transporter proteins found in the OF state at any moment (P(OF)) reflects the fraction of time each transporter spends in OF at steady state (in saturating MgATP) or at equilibrium (in apo condition, in saturating MgATgS, MgAMP-PNP, MgADP, or ATP/EDTA, or with E-to-Q mutation). Thus, P(OF)=t(OF)/(t(IF)+t(OF)), where t(IF) and t(OF) are the average life times of the IF and OF conformations, respectively. E.g., the larger P(OF) in MgATP+Vi than in MgATP observed for WT Tm287/288, or of the E-to-Q mutant compared to WT (both in MgATP), likely reflect longer t(OF) values due to trapping in post- and prehydrolytic OF states, respectively. Similarly, the smaller P(OF) values of WT in MgAMP-PNP, ATP/EDTA, or ATPgS compared to those of the E-to-Q mutant in MgATP likely reflect both shorter t(OF) (due to a less stable NBD dimer) and longer t(IF) (due to less efficient NBD dimerization). Thus, the efficiencies of the nucleotides to stabilize the OF conformation correlates with their structural similarities to ATP, which is the strongest "glue" for holding the NBDs together (but is hydrolyzed in WT). Such an interpretation would be consistent with observations on other ABC proteins. E.g., NBD dimerization is facilitated by an intra-NBD subdomain closure in which the α-helical tail subdomain approaches the β-sheet head subdomain; this "induced fit" is elicited by binding of ATP, but not ADP (Karpowich et al., 2001, Structure 9:571-586). Consequently, isolated E-to-Q mutant NBDs form stable dimers in MgATP, but less so in AMP-PNP, ATPgS, or ATP without Mg^2+^, and not at all in ADP (Moody et al., 2002, J Biol Chem 277:21111-14; Veron et al., 2003, J Mol Biol 334:255-267). Similar conclusions have been reached on CFTR, for which the rates for the IF->OF transition (pore opening) and OF->IF transition (pore closure) can be directly measured, and have been observed in the presence of all of these nucleotides (e.g., Vergani et al., 2003, J Gen Physiol 121:17-36).

6) Temperature-dependent shift in IF->OF conformational distribution

The observed larger P(OF) at 80 °C, compared to 25 °C, for all nucleotides is again interpreted to reflect temperature dependence of nucleotide affinities (subsection “Conformational equilibria influenced by temperature”, fourth paragraph). However, this interpretation cannot be correct, as all nucleotides were applied at saturating concentrations at both temperatures (see comment 4, above). A more likely explanation is a steeper temperature dependence (larger deltaH^+^+) of the IF->OF transition compared to the OF->IF transition, under both hydrolytic and non-hydrolytic conditions. Indeed, this would be consistent with temperature-dependence of CFTR gating transitions (Mathews et al., 1998, J Membrane Biol 163:55-66; Csanady et al., 2006, J Gen Physiol 128:523-533), for which deltaH^+^+ for pore opening (IF->OF transition) is ~117 kJ/mol (Q10=~4.6), while deltaH^+^+ for pore closure (OF->IF transition) is ~69 kJ/mol (Q10=~2.5) under hydrolytic, and ~40 kJ/mol (Q10=~1.7) under non-hydrolytic conditions, causing strong temperature dependence of open probability (i.e., P(OF)).

7) Decoupling of NBD movements at 80 °C in the absence of nucleotides: "disengagement of the NBDs resulting from incubation at high temperature is not propagated to the TMDs and does not result in a further opening of the inward-facing cavity. In other words, the NBDs are decoupled from the rest of the transporter when no nucleotides are present". Strictly speaking this statement seems to apply only to the degenerate site. For the consensus site (Tm287/460-Tm288/363) the appearance of the peak at ~5.6 nm seems to correspond to the appearance of a peak at ~6.2 nm for one of the intracellular TM pairs (Tm288/131-Tm288/248). Could it be that movements of the consensus site remain more tightly coupled to movements of the intracellular ends of the TM helices even at 80 °C, in the absence of nucleotides?

8) Cartoon functional cycle in Figure 10

State 2 in this cartoon is coined as an OF state with a closed NBD dimer, but characterized by "weak nucleotide binding". What is the evidence for the existence of such a state? Indeed, evidence suggests that the IF->OF transition is reversible, but only at a very slow rate. Further, State 3 in the cartoon merges pre- and posthydrolytic OF states into a single compound state. Wouldn't it be more logical to call State 2 the (stable) prehydrolytic, and State 3 the posthydrolytic OF state, with step 2->3 representing ATP hydrolysis? The E-to-Q mutation would then lock the transporter in long-lived State 2, whereas ATP+Vi would stabilize State 3…

---

## [Author Response]

*Essential revisions:*

*1) We feel that ATPase activity measurements for the 6 pairs of labeled Cys residues is not sufficient to conclude the transporter is functional. We request that the authors provide measurements of transport or at least substrate-stimulated ATPase activity for all six pairs of spin-labeled Cys mutants.*

We have experimentally addressed this point by determining Hoechst 33342-stimulated ATPase activities of the six spin-labeled, membrane-reconstituted TM287/288 pairs (new Figure 1—figure supplement 2). The measurements revealed that all spin-labeled Cys mutants remained functional. Interestingly, the drug stimulation patterns of the spin-labeled Cys mutants introduced at the extracellular part of TM287/288 were somewhat different from the wildtype control. In the revised version of the manuscript, we interpret this finding with the extracellular gate being potentially important for drug transport.

We would like to point out that transport data with TM287/288 are difficult to perform *in vitro*, because the activity of the transporter (stemming from a thermophile) is very low at the assay temperatures used with proteoliposomes consisting of *E. coli* lipids and phosphatidylcholine. Reconstitution into native *Thermotoga maritima* lipids would go beyond the scope of this work.

*2) DEER measurements in the presence of Mg-ATP for the 150/295 pair is conspicuously missing from the figures. Could the authors provide this data to complete the dataset?*

This pair showed a big overlap of distances in the IF and OF states, therefore it did not report the changes very accurately, this was the reason why ATP-Mg state had been previously excluded. However, we agree that for the completeness of the results this trace should be added. The DEER experiment was performed and added in Figure 2, Figure 2—figure supplement 1 and Figure 2—figure supplement 2. In the presence of ATP-Mg a broad distance distribution indicative of a large fraction of state was detected.

*3) It is not clear how the range of distance characteristics of the IF and OF conformation as given in Figure 1,Figure 4,Figure 6 and the supplements were achieved. In some cases (e.g. 150/295, 460/363) they clearly deviate from the predicted distances using MMM. Nonetheless, the authors stated that the distances for at least the IF conformation agree with the crystal structure. This is not always true as can be seen in Figure 1—figure supplement 1 and should be clarified and discussed in light of possible alternative conformations.*

Indeed, we did not clarify the existing deviations between experimental and simulated data. The simulation program MMM is based on spin label rotamer libraries, which are obtained by MD simulations at different temperatures. Different libraries (e.g. 2015 library at ambient temperature, or 2013 library at cryogenic temperature) give different shapes of the distance distributions, due to the different weight of each rotamer in the calculation. The comparison between two libraries is presented in Figure 1—figure supplement 1. A 3.5-4 Å rmsd between experimental and calculated mean distances was found with the 2013 rotamer library (Jeschke, G. (2013). Conformational dynamics and distribution of nitroxide spin labels. Progress in Nuclear Magnetic Resonance Spectroscopy 72, 42-60). In the apo state, the simulated mean distances in the six pairs agree within this rmsd to the experimental ones, therefore we stated that our data are in agreement with the X-ray structure. The new library (2015) gives somehow slightly worse results (especially for pairs 150-295 and 350- 475), highlighting the coarse-grained approach used to simulate the distance distribution. We chose to show both libraries to highlight differences between them, an interesting information for the EPR community. For the homology model, the errors due to the coarse-grained library approach add to the intrinsic errors that such models can produce in terms of side chain arrangements. The latter is influencing the number and type of spin- labeled rotamers calculated, therefore the resulting distance distributions. Despite these issues, we observed a remarkable agreement between experimental and simulated distance changes. The 2013 library performs again better, with exception of the pair 460-363. We clarified this in the text. The range of distances in shaded pink and blue are indicative of the experimentally measured distributions in the vanadate-trapped and apo state, respectively and are depicted in the panels to guide the eye. This was added in the legends of Figure 2 and Figure 4.

*4) Competition studies using ATP hydrolysis assay (Figure 3)*

*4.1) Fitting such dose response curves (fractional ATP hydrolysis turnover rates at a fixed [ATP], plotted as a function of competitor concentration) with sums of exponentials is entirely inadequate: no mechanistic model predicts exponential dose response relationships. Instead these curves should be fitted with hyperbolic functions of the form v/v0=IC50/(<I>+IC50), where <I> is the concentration of the competitor, and v0 is the rate measured at <I>=0. Using logarithmic abscissae might also help visualizing these IC50 values.*

*4.2. The IC50 values themselves are not very informative since they depend on the concentration of the substrate (ATP) as follows: IC50=K_i_(1+[S]/K_m_), where K_i_ is the apparent affinity of the inhibitor, [S] is the concentration of the substrate, and K_m_ is the Michaelis constant for ATP hydrolysis in the absence of an inhibitor. (The above equations assume competition for a single site, but such a simplification does not seem inappropriate, since at the very high [ATP] used the high-affinity degenerate site likely remains loaded with ATP.) Thus, given a K_m_ of ~20 μm for ATP (line 376), and [ATP]=500 μm in these assays, the relationship IC50=~26xK_i_ holds. The IC50 values of 44 μm, >800 μm, and 565 μm for ATPgS, AMP-PNP, and ADP (line 200) therefore correspond to K_i_ values (binding affinities) of 1.7 μm, >31 μm, and 22 μm, respectively. In addition to the IC50 values, these calculated K_i_ values should also be provided for each nucleotide, transporter, and temperature, as these are highly relevant for the interpretation of the DEER experiments performed in the presence of those nucleotides alone (see comment 5., below).*

We would like to thank the reviewers for this suggestion regarding data fitting and interpretation. We agree with the assumption that we mainly look at the competition for the consensus site here. However, as outlined in the manuscript, the reality is likely more complex. Nucleotides will have different affinities for the degenerate site as well and likely exhibit different capabilities for the closure of the degenerate site by interacting with the non-canonical ABC signature motif of the opposite monomer. A further complication is the allosteric coupling between degenerate and consensus site.

We have now fitted the inhibition data with a hyperbolic function. Because inhibition was not complete even at high inhibitor concentration, the given formula *v/v0=IC50/([Inhibitor]+IC50)* needed to be extended. We used the formula *f = y0+(a*b)/(b+x)*, in which f is v/v0, y0 is the residual activity at infinite [Inhibitor], a is the maximal degree of inhibition (a + y0 = 100% ), x is [Inhibitor] and b is the IC50.

K_i_ values were calculated as suggested with the formula *IC50=Ki(1+[S]/K_m_).* To this end, accurate K_m_ values needed to be determined for TM287/288 and BmrCD at the respective temperatures. For TM287/288 with its high apparent affinity for ATP, it was challenging to determine K_m_ values using the malachite green detection method; ATP turnover had to remain small (< 20% ) even at the lowest ATP concentration resulting in small amounts of liberated Pi at these concentrations for detection. Therefore, the error bars are somewhat higher for these measurement points for technical reasons.

To verify the correctness of our assumptions (i.e. to verify the relation *IC50=K_i_(1+[S]/K_m_*) at least for one example, K_m_ and v_max_ values for ATP hydrolysis at different ATP-g-S concentrations were determined at 50 °C for TM287/288. K_m_/v_max_ was plotted against the inhibitor concentration, the values were fitted by linear regression and the intercept with the x-axis was used to determine K_i_. As can be seen in the graph plotted below, the determined data points could be fitted well by linear regression. The determined K_i_ for ATP-g-S of

43.5 nM was basically identical with the procedure described above (IC50 determination by hyperbolic function and IC50 to K_i_ conversion by formula *IC50=K_i_(1+[S]/K_m_)*, for which a K_i_ for ATP-g-S of 43.9 nM was determined (see Table 2). Because we only have determined the K_i_ value for ATP-g-S at 50 °C in this manner, we did not include this piece of data in the paper.

Author response image 1.**DOI:**
http://dx.doi.org/10.7554/eLife.20236.027

*5) Interpretation of nucleotide efficiencies to stabilize OF state*

*Efficiencies of various nucleotides to stabilize the OF state are repeatedly interpreted to reflect the affinities of the respective nucleotides (subsection “Conformational equilibria explored in ATPase deficient E-to-Q mutant of TM287/288”, subsection “Conformational equilibria influenced by temperature”, fourth paragraph and Discussion, fifth paragraph). However, this interpretation is incorrect, since – except for the apo condition – the DEER measurements were all performed at saturating nucleotide concentrations (2.5 mM, whereas K_i_ values are low μM for each; see comment 4., above). Instead, the fraction of transporter proteins found in the OF state at any moment (P(OF)) reflects the fraction of time each transporter spends in OF at steady state (in saturating MgATP) or at equilibrium (in apo condition, in saturating MgATgS, MgAMP-PNP, MgADP, or ATP/EDTA, or with E-to-Q mutation). Thus, P(OF)=t(OF)/(t(IF)+t(OF)), where t(IF) and t(OF) are the average life times of the IF and OF conformations, respectively. E.g., the larger P(OF) in MgATP+Vi than in MgATP observed for WT Tm287/288, or of the E-to-Q mutant compared to WT (both in MgATP), likely reflect longer t(OF) values due to trapping in post- and prehydrolytic OF states, respectively. Similarly, the smaller P(OF) values of WT in MgAMP-PNP, ATP/EDTA, or ATPgS compared to those of the E-to-Q mutant in MgATP likely reflect both shorter t(OF) (due to a less stable NBD dimer) and longer t(IF) (due to less efficient NBD dimerization). Thus, the efficiencies of the nucleotides to stabilize the OF conformation correlates with their structural similarities to ATP, which is the strongest "glue" for holding the NBDs together (but is hydrolyzed in WT). Such an interpretation would be consistent with observations on other ABC proteins. E.g., NBD dimerization is facilitated by an intra-NBD subdomain closure in which the α-helical tail subdomain approaches the β-sheet head subdomain; this "induced fit" is elicited by binding of ATP, but not ADP (Karpowich et al., 2001, Structure 9:571-586). Consequently, isolated E-to-Q mutant NBDs form stable dimers in MgATP, but less so in AMP-PNP, ATPgS, or ATP without Mg^2+^, and not at all in ADP (Moody et al., 2002, J Biol Chem 277:21111-14; Veron et al., 2003, J Mol Biol 334:255-267). Similar conclusions have been reached on CFTR, for which the rates for the IF->OF transition (pore opening) and OF->IF transition (pore closure) can be directly measured, and have been observed in the presence of all of these nucleotides (e.g., Vergani et al., 2003, J Gen Physiol 121:17-36).*

We thank the reviewers for bringing up this important issue. Indeed, all measurements were carried out at nucleotide concentrations of at least 7 fold above the K_m_ (for ATP) and 90 fold above K_i_ (for the analogs, ADP and vanadate), i.e. the results clearly represent the situation under saturating conditions. We found an intriguing correlation between lower K_i_ values and higher OF fractions in our data. However, due to the saturating conditions we cannot argue (as we did in the initial version of the manuscript) with different nucleotide affinities being responsible for the differences regarding the population of the OF state. This also applies to the EtoQ mutant.

A notable exception may have been ATP-EDTA, for which no K_m_ nor K_i_ could be determined due to lack of ATP hydrolysis. To experimentally address the question whether ATP-EDTA was added at saturating concentrations, we performed additional experiments at 14 mM ATP + 2.5 mM EDTA for TM287/288 and obtained indistinguishable distance distribution with respect to the 2.5 mM ATP-EDTA conditions. Both conditions showed fractions of IF and OF states. We therefore concluded that the ATP-EDTA measurements for TM287/288 were conducted under saturating conditions. We also used 10 mM ATP concentration for BmrCD in the presence of EDTA or Mg and found that the distance distributions were indistinguishable to those obtained with 2.5 mM nucleotide. Due to the absence of an IF fraction in the presence of EDTA, we cannot conclude that we were at saturating conditions at 10 mM ATP-EDTA. However, the absence of even a small fraction corresponding to the OF distance points to either a distinct energy landscape for BmrCD with respect to TM287/288 or to an extremely weak affinity of ATP in the absence of Magnesium ions (> 100 mM).

In this context one should keep in mind that the K_i_ values were determined as a measure to inhibit ATP hydrolysis. The measurements therefore do provide direct insight into their affinities for the asymmetric nucleotide binding sites. For ATP to be hydrolyzed (which is the readout of our assay), the consensus site (and most probably also the degenerate site) need to close. For a nucleotide analog it is for example sufficient to compete for ATP binding at the consensus site to cause inhibition without binding to the degenerate site. When the same nucleotide analog is then added alone to the transporter, it may act as a “lousy glue” to bridge Walker A with the opposite signature motif (this can happen at the degenerate or the consensus site).

What we observed in our experiments is that high fractions of the OF state were correlated with the ability of the nucleotide to compete strongly with ATP hydrolysis. Thus, strong competition in the ATPase assay appears to be correlated with the structural similarity between the analog and ATP and we agree with the reviewers’ interpretation that “the efficiencies of the nucleotides to stabilize the OF conformation correlates with their structural similarities to ATP, which is the strongest "glue" for holding the NBDs together (but is hydrolyzed in WT)”. Of note, hydrolyzable nucleotides (ATP and ATPγS) act as strong “glues” whereas non-hydrolyzable ones (ADP and AMP-PNP) don’t. That’s why we propose in this study (also based on previous literature by other authors) that when a nucleotide is committed to hydrolysis, NBD dimer stability is strongly increased (see also comments below on the mechanism). Hence, we prefer to explain the low fraction of OF state in the presence of saturating concentrations of AMP-PNP-Mg or ADP-Mg with less efficient stabilization of the NBD dimer by these nucleotides. The reviewers bring up the alternative explanation of less efficient NBD dimerization and provide literature suggestions on the dimerization of isolated NBDs and the channel opening of CFTR on this topic. NBD closure and NBD stabilization are difficult to discern experimentally, because there are (at least to our knowledge) no kinetic data as for example measured by surface plasmon resonance which directly looked at NBD association and dissociation to determine k_on_ (i.e. rate of NBD closure) and k_off_ (rate of NBD dissociation providing insights into the stability of the NBD dimer). Current knowledge on this topic is based on crystal structures and size exclusion chromatography analyses, which provide some information about the affinity constants of closed NBDs, but no information about the rates of association and dissociation. The studies on CFTR are very interesting in this context, in which the rate of channel opening is used as readout to look at NBD closure. However, conformational changes at the NBDs are not transmitted immediately to the channel portion at the TMDs, so that the readout of channel opening only indirectly reports on the NBD closure.

We have rephrased the sections in the manuscript where nucleotide affinities were used to explain differences in the population of the OF state.

*6) Temperature-dependent shift in IF->OF conformational distribution*

*The observed larger P(OF) at 80 °C, compared to 25 °C, for all nucleotides is again interpreted to reflect temperature dependence of nucleotide affinities (subsection “Conformational equilibria influenced by temperature”, fourth paragraph). However, this interpretation cannot be correct, as all nucleotides were applied at saturating concentrations at both temperatures (see comment 4, above). A more likely explanation is a steeper temperature dependence (larger deltaH^+^+) of the IF->OF transition compared to the OF->IF transition, under both hydrolytic and non-hydrolytic conditions. Indeed, this would be consistent with temperature-dependence of CFTR gating transitions (Mathews et al., 1998, J Membrane Biol 163:55-66; Csanady et al., 2006, J Gen Physiol 128:523-533), for which deltaH^+^+ for pore opening (IF->OF transition) is ~117 kJ/mol (Q10=~4.6), while deltaH^+^+ for pore closure (OF->IF transition) is ~69 kJ/mol (Q10=~2.5) under hydrolytic, and ~40 kJ/mol (Q10=~1.7) under non-hydrolytic conditions, causing strong temperature dependence of open probability (i.e., P(OF)).*

This is an interesting comment. Although we observed lower K_i_ values for AMP-PNP and ATP-g-S at 50 °C compared to 25 °C, they cannot explain the changes in the IF/OF ratios observed at higher temperature, because in any case the nucleotides have been added at saturating concentrations. Although we do not provide direct experimental evidence for TM287/288, it is plausible to explain our observations with a steeper temperature dependence for the IF->OF transition compared to the OF-> IF transition, resulting in an overall shift to the OF state under equilibrium or steady-state conditions as applied in the context of our measurements.

Statements regarding K_i_ changes of AMP-PNP and ATP-g-S being the reason behind the increased population of the OF state at high temperatures were removed and different steepness of temperature dependence of the transitions was included as a plausible explanation of our results.

*7) Decoupling of NBD movements at 80 °C in the absence of nucleotides: "disengagement of the NBDs resulting from incubation at high temperature is not propagated to the TMDs and does not result in a further opening of the inward-facing cavity. In other words, the NBDs are decoupled from the rest of the transporter when no nucleotides are present". Strictly speaking this statement seems to apply only to the degenerate site. For the consensus site (Tm287/460-Tm288/363) the appearance of the peak at ~5.6 nm seems to correspond to the appearance of a peak at ~6.2 nm for one of the intracellular TM pairs (Tm288/131-Tm288/248). Could it be that movements of the consensus site remain more tightly coupled to movements of the intracellular ends of the TM helices even at 80 °C, in the absence of nucleotides?*

Thanks for valuable comment. Please note that a known drawback in using Tikhonov regularization is that a compromise between the smoothness of the distance distribution and the quality of form factor fit needs to be made to yield a reliable distance distribution. Thus, very often – especially when several distances with different width are present – higher fitting quality is preferred. In such cases, peaks with height smaller than 10% of the highest peak (when normalized to the area under the distribution curve) are considered to arise from noise and therefore ignored. The mentioned ~6.2 nm in pair 131^TM288^/248 ^TM288^ has a height which is smaller than 10% of the main peak and cannot be reliably used to draw conclusions. Instead, ~5.6 nm peak in pair 460 ^TM287^/363 ^TM288^ along with ~8 nm peak further shows the disengaged consensus NBD site similar to what is seen in degenerate site at high temperature.

8) Cartoon functional cycle in Figure 10

State 2 in this cartoon is coined as an OF state with a closed NBD dimer, but characterized by "weak nucleotide binding". What is the evidence for the existence of such a state? Indeed, evidence suggests that the IF->OF transition is reversible, but only at a very slow rate. Further, State 3 in the cartoon merges pre- and posthydrolytic OF states into a single compound state. Wouldn't it be more logical to call State 2 the (stable) prehydrolytic, and State 3 the posthydrolytic OF state, with step 2->3 representing ATP hydrolysis? The E-to-Q mutation would then lock the transporter in long-lived State 2, whereas ATP+Vi would stabilize State 3.

We agree with the reviewers’ constructive assessment on transport cycle and the cycle has been modified to accommodate separate pre- and post-hydrolytic states and the effect of E-to-Q mutation (inhibiting the transition from pre-to-post hydrolysis) as well as trapping with vanadate (stabilization of the post-hydrolysis state).

Our model also accounts for the fact that hydrolyzable nucleotides act as strong glue to keep the NBDs together (i.e. ATP and ATPγS). Our data suggest (as many other data of the ABC transporter field) that when a nucleotide is committed to hydrolysis, this increases the affinity of the nucleotide and in turn strongly stabilizes the NBD dimer. As our experiments show, strong stabilization can be achieved in the E-to-Q mutant and ATP- Mg, without hydrolysis of the nucleotide or using vanadate trapping of ATP. On the other hand, ATP-EDTA (which is not committed to ATP hydrolysis), as well as non-hydrolyzable AMP-PNP, are much less capable of NBD-stabilization. We think that the NBDs first close (mainly driven by Brownian motion) and that during closure, two nucleotides bind. These nucleotides stabilize the closed NBD dimer at the end of the trajectory, acting as molecular glue. In case of ATP-EDTA, the glue appears weak and the OF state is poorly populated. In case of AMP-PNP, the glue is even too weak to populate any detectable OF state, at least in wildtype TM287/288. This is the state 2 we speak about in our model; it is a state in which the NBDs sandwich two nucleotides in a reversible fashion. After this initial closure of the NBDs, the nucleotide at the consensus site is either committed to hydrolysis (ATP or ATP-g-S), or not (AMP-PNP, ADP or ATP-EDTA). Only hydrolysable nucleotides lead to high fractions of OF state. In case of BmrCD, only ATP-Mg leads to any detectable OF state. We explain this by an irreversible commitment to hydrolysis leading to a strong NBD dimer, which only can be released again by ATP hydrolysis.